# TET1 is a beige adipocyte-selective epigenetic suppressor of thermogenesis

Sneha Damal Villivalam[1], Dongjoo You[1], Jinse Kim[1], Hee Woong Lim [2], Han Xiao[1], Pete-James H. Zushin[1], Yasuo Oguri[3], Pouya Amin[1] & Sona Kang [1✉]

It has been suggested that beige fat thermogenesis is tightly controlled by epigenetic regulators that sense environmental cues such as temperature. Here, we report that subcutaneous adipose expression of the DNA demethylase TET1 is suppressed by cold and other stimulators of beige adipocyte thermogenesis. TET1 acts as an autonomous repressor of key thermogenic genes, including *Ucp1* and *Ppargc1a*, in beige adipocytes. Adipose-selective *Tet1* knockout mice generated by using Fabp4-Cre improves cold tolerance and increases energy expenditure and protects against diet-induced obesity and insulin resistance. Moreover, the suppressive role of TET1 in the thermogenic gene regulation of beige adipocytes is largely DNA demethylase-independent. Rather, TET1 coordinates with HDAC1 to mediate the epigenetic changes to suppress thermogenic gene transcription. Taken together, TET1 is a potent beige-selective epigenetic breaker of the thermogenic gene program. Our findings may lead to a therapeutic strategy to increase energy expenditure in obesity and related metabolic disorders.

[1] Nutritional Sciences and Toxicology Department, University of California Berkeley, Berkeley, CA 94720, USA. [2] Division of Biomedical Informatics, Cincinnati Children's Hospital Medical Center Department of Pediatrics & Biomedical Informatics, University of Cincinnati, 3333 Burnet Ave. MLC 7024, Cincinnati, OH 45229, USA. [3] UCSF Diabetes Center, Eli and Edythe Broad Center of Regeneration Medicine and Stem Cell Research, Department of Cell and Tissue Biology, University of California, San Francisco, CA 94143, USA. ✉email: kangs@berkeley.edu

Mammals have at least two types of thermogenic adipocytes, brown and beige, that play a central role in regulating energy homeostasis[1,2]. In rodents, classical brown adipose tissue (BAT) exists in discrete anatomical depots, such as the interscapular regions, while beige adipocytes sporadically arise within white adipose tissue (WAT)[1,2]. These two thermogenic adipocyte subtypes are similar in multiple respects: both contain multilocular lipid droplets, have a high mitochondrial content, and express key thermogenic genes such as *Ucp1*, *Ppargc1a*, and *Cidea*[1,2]. At the same time, beige and brown adipocytes have several distinct characteristics that distinguish them as two different cell types. For instance, their developmental origins are different. While classical brown adipocytes derive from Myf5-positive precursors during embryonic development, beige adipocytes postnatally develop in the WAT depots of adults and derive from multiple origins depending on the depot[3–7]. In addition, molecular profiling studies highlight significant differences between the gene signatures of brown vs. beige adipocytes[8,9]. Lastly, the plasticity of thermogenic activity remarkably differs between these two cell types. Brown adipocytes are constitutively active and express high levels of UCP1 and other thermogenic genes. Their thermogenic activity can be further increased to a moderate degree upon stimulation. On the other hand, beige adipocytes express very low levels of thermogenic genes but robustly induce their expression in response to external stimuli, thus displaying a greater degree of thermogenic plasticity[1,2].

Beige fat formation, also called "browning" or "beiging", is induced by various environmental cues, including cold exposure, exercise, and PPARγ agonist[1,2], and it results in the production of heat by burning stored fat. Conversely, beige adipocytes also undergo "whitening", returning them to a white adipocyte–like phenotype, in response to thermoneutrality[10], impaired β-adrenergic signaling[11], triglyceride hydrolase deficiency[12], and other cues[13]. In mice, beige fat contains one tenth the amount of UCP1, a key thermogenic protein, than brown fat;[14] however, the total amount of beige fat can be greater than brown fat and can have a bigger impact on energy homeostasis, as it can be recruited en masse in many white depots[14]. Notably, recent studies have identified additional mechanisms through which beige fat modulates whole-body metabolism such as creatine cycling[15] and Serca2b-mediated calcium cycling[16]. In humans, although still debatable, a substantial number of studies suggest that thermogenic adipocytes are recruited upon cold acclimatization[17,18] leading to increased energy expenditure and a beneficial impact on glucose metabolism[18,19]. Due to this remarkable plasticity and the relevance to human obesity, beige adipocytes are an attractive therapeutic target for obesity and related metabolic diseases.

The browning and whitening of beige and brown fat involves dramatic changes in morphology, transcription, and chromatin landscape[10,20–25]. Therefore, epigenetic regulators are likely to play a key role in this process. The ten-eleven translocation (TET) proteins (TET1, 2, and 3) are the enzymes that oxidize methylated cytosine. In addition to participating in the initial step of DNA demethylation, TETs play versatile roles in transcription regulation[26] by acting as both transcriptional co-activators and co-repressors[26]. Notably, the transcriptional regulation activity of TET proteins can be dependent or independent of their demethylase activity through interacting with other transcriptional regulators and chromatin modifiers[27–31].

Here, we report that TET1 expression in subcutaneous adipose tissue shows reciprocal regulation with UCP1 in response to ambient temperature changes and cAMP signaling. We demonstrate that *Tet1* loss-of-function leads to cell-autonomous increases in cAMP-induced expression of thermogenic genes, including *Ucp1*, and increases in mitochondrial respiration in beige adipocytes. Consistent with this, conditional deletion of *Tet1* in adipose tissue increases browning of subcutaneous fat, leading to reduced adiposity and improved cold tolerance, glucose tolerance, and insulin sensitivity. Moreover, the knockout mice are protected from diet-induced obesity and metabolic impairment. Mechanistically, we identify that TET1 collaborates with HDAC1 to mediate the epigenetic changes that suppress thermogenic gene transcription in a DNA demethylase-independent manner. Together, our results suggest that TET1 is a potent epigenetic sensor of ambient temperature and modulates the temperature-mediated browning of beige adipocytes.

## Results

**Adipose TET1 levels in response to temperature and cAMP.** First, we compared the expression of *Tets* across tissues and noted that all three *Tets* are moderately expressed in inguinal and epididymal white adipose tissues (iWAT and eWAT) and less expressed in BAT (Fig. 1a–d). To identify a potential role for the DNA methylation machinery in the temperature-induced plasticity of beige and brown adipocytes, we examined how TET expression in adipose tissue is affected by changes in ambient temperature. Inguinal white adipose tissue (iWAT), epididymal WAT (eWAT), interscapular brown fat tissue (BAT) were taken from wild-type C57BL/6J mice that were housed at RT (23 °C) before being subjected to cold (4 °C) or thermoneutral (TN, 30 °C) temperatures for 24 h. As expected, *Ucp1* levels overall went up in all three depots when mice were housed for 24 h at cold (4 °C) and down when housed at thermoneutrality (TN, 30 °C) (Fig. 1e, Supplementary Fig. 1a, e). Importantly, the expression of *Tet1* was most dramatically regulated by the changes in ambient temperature, especially in iWAT (Fig. 1f). Heat increased the mRNA expression of *Tet1* by ~20 fold whereas cold reduced it by ~60% (Fig. 1f). The temperature-sensitive regulation of TET1 in iWAT was also confirmed by western blotting (Fig. 1f). A similar expression pattern was observed in BAT and eWAT albeit smaller in magnitude (Supplementary Fig. 1b, f).

Similar to *Tet1*, the expression of *Tet2* was also changed but to a lesser degree in iWAT (Fig. 1g, Supplementary Fig. 1c, g). In contrast, *Tet3* levels did not show consistent changes between depots; they had a ~3 fold increase in iWAT (Fig. 1h), a decrease in BAT (Supplementary Fig. 1d), and no change in eWAT (Supplementary Fig. 1h) under cold conditions. The anti-correlation between *Tet1* levels with *Ucp1* in vivo prompted us to determine their expression levels in three different shades of adipocyte cell lines: 3T3-L1 cells (considered "white") and immortalized "beige" and "brown" adipocytes. As expected, UCP1 protein expression was not detectable in mature 3T3-L1 adipocytes, had intermediate expression in mature beige adipocytes, and high expression in mature brown adipocytes (Fig. 1l). Interestingly, TET1 expression was increased during white adipogenesis and reduced during beige and brown adipogenesis, displaying an anti-correlation with UCP1 levels (Fig. 1i, l). On the other hand, both the expression of TET2 and TET3 was overall higher in all three types of mature adipocytes compared to that of preadipocytes (Fig. 1j–l). As we noted that *Tet1* expression was relatively higher in iWAT compared to BAT (Fig. 1a, d), we looked into how the regulation of *Tet1* expression is regulated in response to thermogenic stimulators (cAMP-inducing forskolin and norepinephrine) in in vitro-differentiated primary inguinal adipocytes. Consistent with in vivo, *Tet1* mRNA expression was reciprocally regulated with *Ucp1* expression in response to the thermogenic stimulators (Fig. 1m, n, Supplementary Fig. 1i, j). Together, our expression data suggested that TET1 and TET2 have a functional role in the regulation of thermogenesis in beige fat, thus we sought to test the effects of their downregulation in beige adipocyte thermogenesis.

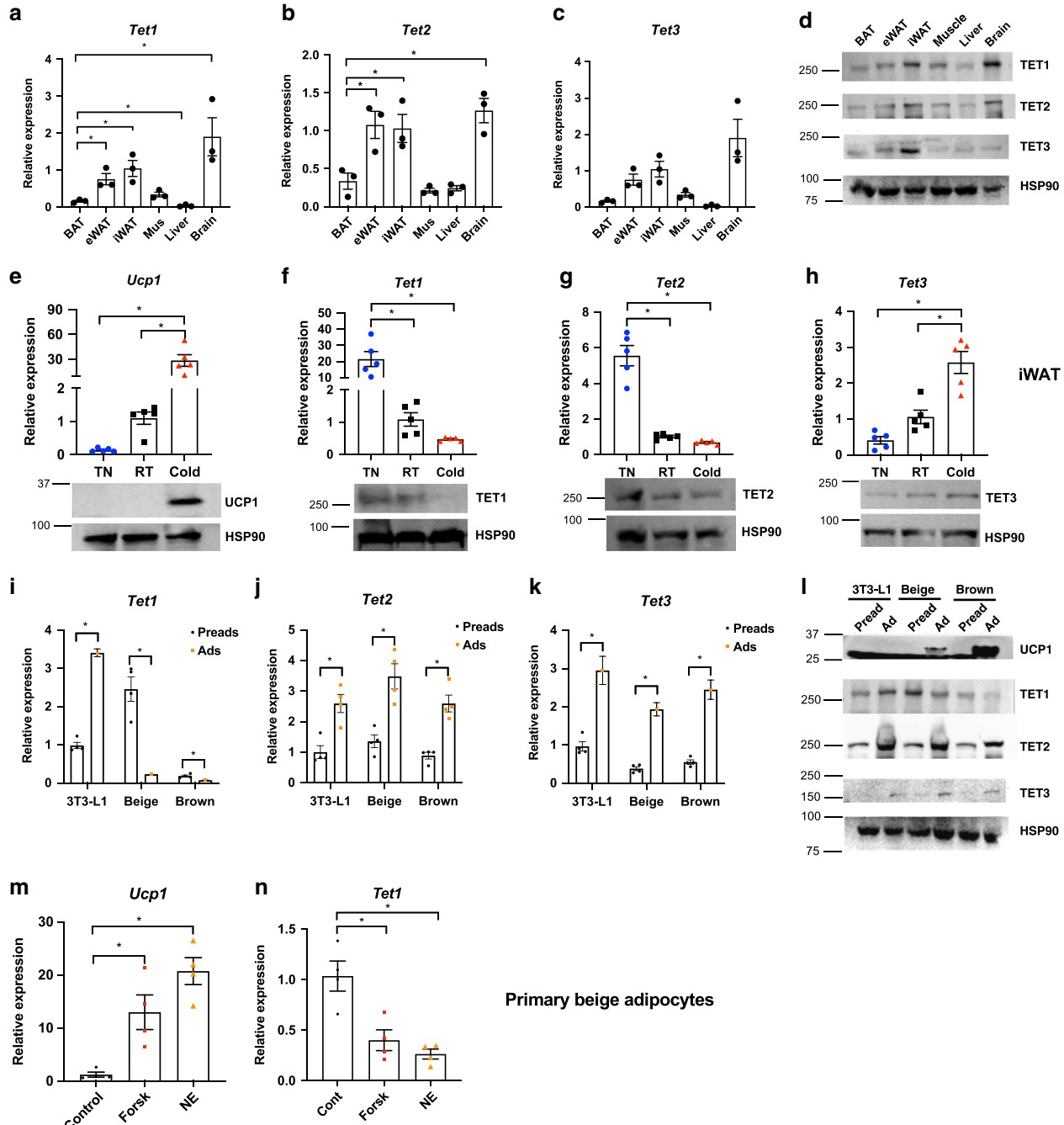

**Fig. 1 Subcutaneous adipose *Tet1* expression is regulated by ambient temperature and cAMP signaling. a–d** Various tissues from wild-type C57BL/6 J mice housed at room temperature (RT) were collected to measure *Tets* mRNA (**a–c**) and protein expression **d** (**a–c**); $n = 3$ per group. Data are expressed as means ± SEM. *denotes $p < 0.05$, determined by two-tailed Student's $t$ test and one-way ANOVA). **e–h** *Tet* and *Ucp1* mRNA and protein expression in iWAT from wild-type male C57BL/6J mice housed at room temperature (RT) and exposed to cold or thermoneutrality (TN) for 24 h ($n = 5$ per group. Data are expressed as means ± SEM. *denotes $p < 0.05$, determined by two-tailed Student's $t$ test and one-way ANOVA followed by Bonferroni post-hoc testing). **i–l** *Tets* mRNA and protein expression were measured in 3T3-L1 and immortalized beige and brown preadipocytes at confluence and after differentiation (**i–k**; $n = 4$ per group. Data are expressed as means ± SEM. *denotes $p < 0.05$, determined by two-tailed Student's $t$ test). **m, n** *Ucp1* and *Tet1* mRNA expression with and without 1 μM forskolin (Forsk) or 1 μM norepinephrine (NE) stimulation for 3 h in mature primary beige adipocytes ($n = 4$ per group, Data are expressed as means ± SEM. *denotes $p < 0.05$, determined by two-tailed Student's $t$ test and one-way ANOVA followed by Bonferroni post-hoc testing). Source data are provided as a source data file.

**TET1 suppresses the thermogenic activation of beige adipocytes.** To test the cell-autonomous role of TETs in the regulation of thermogenic genes, we performed gain- and loss-of-function studies of individual TETs using an immortalized beige cell line. Since TETs are pro-adipogenic[32] we conducted the studies in fully mature beige adipocytes using short hairpin RNAs (shRNAs) (Supplementary Fig. 2a–c) to avoid any effect on differentiation. Remarkably, knockdown of *Tet1*, but not *Tet2* or *Tet3*, enhanced the forskolin-stimulated expression of *Ucp1* and some of key thermogenic markers including *Ppargc1a*, *Cidea*, and *Elovl3* (Fig. 2a–d). Notably, there was no change in the expression levels of general adipocyte markers such as *Pparg* and *Fabp4* (Fig. 2e, f).

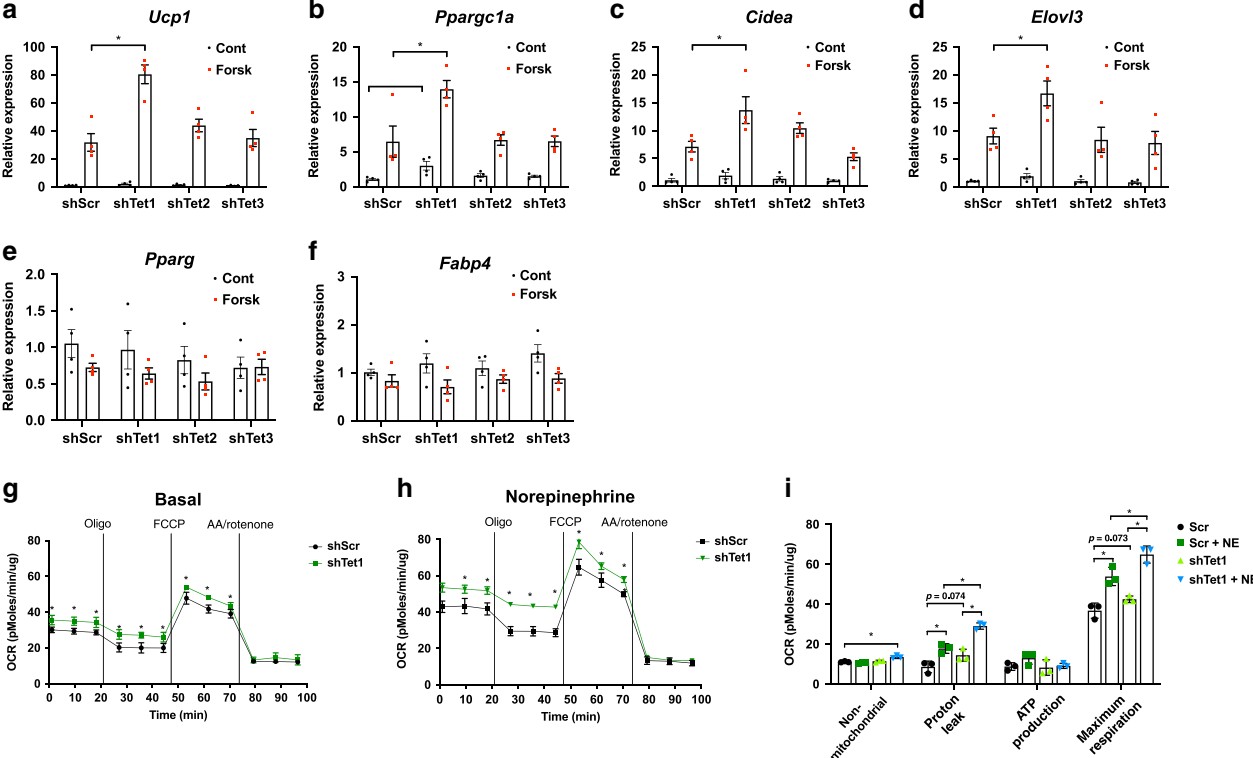

**Fig. 2 Tet1 loss-of-function in vitro increases thermogenesis in beige adipocytes. a–f** Differentiated beige adipocytes were transduced with hairpins against *Tet1, 2,* and *3* and scrambled control shRNA (shScr). The basal- and forskolin-stimulated levels of key adipocyte thermogenic gene transcripts were measured by qPCR ($n = 4$ per group. Data are expressed as means ± SEM. *denotes $p < 0.05$, determined by two-tailed Student's *t* test and two-way ANOVA followed by Bonferroni post-hoc testing). **g–i** Basal and norepinephrine (NE)-stimulated mitochondrial respiration under various drug treatments was measured in *Tet1* knockdown and scramble beige adipocytes ($n = 3$ per group. Data are expressed as means ± SEM. *denotes $p < 0.05$, determined by two-tailed Student's *t* test and one-way ANOVA). (Oligo; Oligomycin, AA; antimycin A). **i** Shown are the various components of oxygen consumption rates with and without NE stimulation from (**g, h**). ($n = 3$ per group. Data are expressed as means ± SEM. *denotes $p < 0.05$, determined by two-tailed Student's *t* test and two-way ANOVA followed by Bonferroni post-hoc testing). Source data are provided as a source data file.

*Tet1* knockdown in beige adipocytes increased the rate of mitochondrial respiration in the presence and absence of norepinephrine (Fig. 2g–i). Conversely, lentiviral overexpression of *Tet1* in mature beige adipocytes (Supplementary Fig. 2d, e) suppressed the expression of thermogenic genes (Fig. 3a–d) without altering the expression of general adipocyte markers (*Pparg* and *Fabp4*) (Fig. 3e, f) and suppressed mitochondrial respiration in the presence and the absence of norepinephrine (Fig. 3g–i). Together, our results suggest that TET1 suppresses adipocyte thermogenesis in a cell-autonomous manner in beige adipocytes.

**Tet1 KO mice display increased cold tolerance and energy expenditure.** To determine the in vivo role of *Tet1* in the regulation of adipose plasticity and the thermogenic gene program, we initially generated adipose-selective *Tet1* knockout mice (AdipoQ-Tet1KO) using adiponectin-Cre mice[33]. However, these knockout mice had poor knockdown efficiency (~10%) even when specifically assessing adipocytes (Supplementary Fig. 3a, b). Therefore, we used Fabp4-Cre[34] to generate an adipose-selective knockout (Adi-Tet1KO) mouse, in which *Tet1* mRNA levels were knocked down by more than 80% in the purified adipocyte fraction obtained from iWAT, eWAT, and BAT (Supplementary Fig. 3c, d). Since there is concern over the target specificity of Fabp4-Cre[35], we thoroughly examined *Tet1* knockdown in multiple other tissues. Other than 12% less expression in the brain, we did not find any significant changes in *Tet1* expression in non-adipose tissues including the stromal-vascular fraction

(Supplementary Fig. 3c, d), which contains macrophages and precursors.

To examine whether TET1 is necessary for temperature-mediated adipose plasticity, we exposed Adi-Tet1KO and WT mice to different ambient temperatures. To assess their cold tolerance, we monitored the rectal temperature of individual mice placed into a 4 °C cold chamber. Adi-Tet1KO mice maintained their core body temperature better than WT mice under cold conditions (Fig. 4a). Consistent with the increased tolerance to cold, the KO mice had a significant increase in oxygen consumption upon cold exposure and a trend towards increased oxygen consumption at RT and TN, as compared to controls (Fig. 4b, c). In addition, the serum release of glycerol was elevated in KO mice upon cold exposure (Fig. 4d), likely due to increased lipolysis to provide fuel for the increased adaptive thermogenesis. There were no changes in food intake, physical activity, or respiratory exchange ratio (RER) between genotypes (Supplementary Fig. 4a–c). We also noted that KO iWAT had more multilocular brown-like adipocytes after 7 days of cold exposure (Fig. 4e), whereas there was no marked difference between WT and KO BAT (Fig. 4e). Also notably, the increased rates of basal and norepinephrine-stimulated respiration was more pronounced in the KO iWAT than in the KO BAT (Fig. 4f, Supplementary Fig. 5a). This depot-biased effect was also observed at molecular levels; increased expression of UCP1 and PGC1α was found in iWAT but not in BAT (Fig. 4g, Supplementary Fig. 5b). This suggests that TET1 is acting selectively on beige adipocytes, which are present in iWAT.

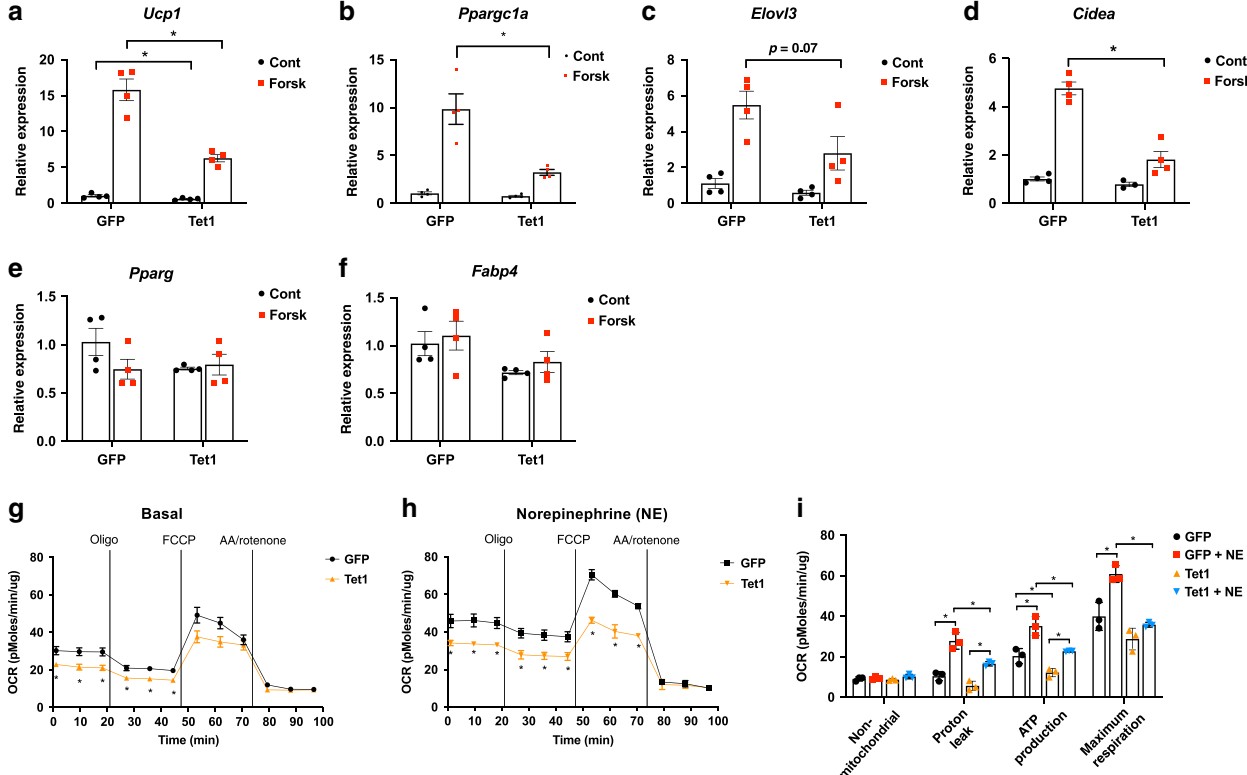

**Fig. 3 Tet1 gain-of-function in vitro suppresses thermogenesis in beige adipocytes. a–f** Differentiated beige adipocytes were transduced with lentiviral expression plasmids for *Tet1* and GFP. The basal- and forskolin-stimulated levels of key adipocyte thermogenic gene transcripts were measured by qPCR ($n = 4$ per group. Data are expressed as means ± SEM. *denotes $p < 0.05$, determined by two-tailed Student's *t* test and two-way ANOVA followed by Bonferroni post-hoc testing). **g, h** Basal and norepinephrine (NE)-stimulated mitochondrial respiration under various drug treatments was measured in *Tet1* overexpression and GFP beige adipocytes ($n = 3$ per group. Data are expressed as means ± SEM. *denotes $p < 0.05$, determined by two-tailed Student's *t* test and one-way ANOVA). (Oligo; Oligomycin, AA; antimycin A). **i** Shown are the various components of oxygen consumption rates with and without NE stimulation from **g, h**. ($n = 3$ per group. Data are expressed as means ± SEM. *denotes $p < 0.05$, determined by two-tailed Student's *t* test and two-way ANOVA followed by Bonferroni post-hoc testing). Source data are provided as a source data file.

Since Fapb4-Cre is also expressed in non-adipose tissues, we did a series of experiments to exclude the possibility of contributing factors from other cell types. It has been proposed that alternative M2 macrophage polarization can induce white adipose tissue browning[36]. Since Fabp4-Cre is active in macrophages, we tested whether there is a contribution from macrophages to the browning phenotype in the KO mice. First, we conducted FACS analysis, finding no major difference in M1 vs. M2 macrophage populations (Supplementary Fig. 6a). Second, we isolated primary macrophages from WT and KO mice and co-cultured them with wild-type beige adipocytes using a transwell system (Supplementary Fig. 6b). We did not find any discernable changes in the expression of thermogenic genes between genotypes (Supplementary Fig. 6c).

PDGFRa-Cre is often used to target the adipocyte precursor population, and indeed, we were able to see ~60% knockdown in the SVF population, defined by the absence of Lin markers (CD31, CD45, Ter119) (Supplementary Fig. 7a). Additionally, PDGFRa is expressed in selected cell types in a wide array of mesenchymal tissues, including the lung, heart, intestine, skin, and cranial facial mesenchyme as well as some adipocyte precursor cells[37–39]. We generated *Tet1* KO mice using PDGFRa-Cre and found no remarkable differences in the thermogenic gene expression of iWAT and BAT in response to β3-agonist or CL316,243 between genotypes (Supplementary Fig. 7b, c). Moreover, PDGFRa-Tet1 KO mice did not show any noticeable phenotypic changes in body weight (Supplementary Fig. 7d, h), body composition (Supplementary Fig. 7e, i), or

oxygen consumption (Supplementary Fig. 7f, g) on both chow and HFD. In accordance with this, PDGFRa-Tet1 KO mice did not display major differences in insulin tolerance and glucose tolerance tests on chow and HFD (Supplementary Fig. 7j–l).

Last, we knocked down *Tet1* expression selectively in inguinal adipose by using adeno-associated virus serotype 8 (AAV8)-Cre, whose expression is driven by the human adiponectin promoter[40,41]. To increase target specificity, we directly injected viruses into the inguinal fat pads of *Tet1*f/f mice. As a result, *Tet1* expression was ~ 60% knocked down in iWAT but not in other tissues including eWAT and BAT (Supplementary Fig. 8a). To assess thermogenesis, AAV8-treated mice were treated with CL316,243, which resulted in increased signs of browning of iWAT at histological and molecular levels compared to control AAV8-treated mice (Supplementary Fig. 8b–e). Moreover, AAV8-hAdi-Cre-treated mice showed increased tolerance to acute cold exposure (Supplementary Fig. 8f). Taken together, our results largely support that TET1 acts as a suppressor of browning in inguinal adipose tissue.

**TET1 acts as a repressor of the thermogenic gene program.** To investigate the mechanism of how TET1 suppresses the thermogenic gene program, we used RNA-Seq to identify the adipocyte-specific target genes of TET1. In short, we profiled fractionated inguinal adipocytes from WT and KO mice held at RT and exposed to cold for 4-hr (Fig. 4h–n, Supplementary Fig. 9a–c). We found that 51 genes were significantly upregulated and 73 genes were downregulated in the KO inguinal adipocytes

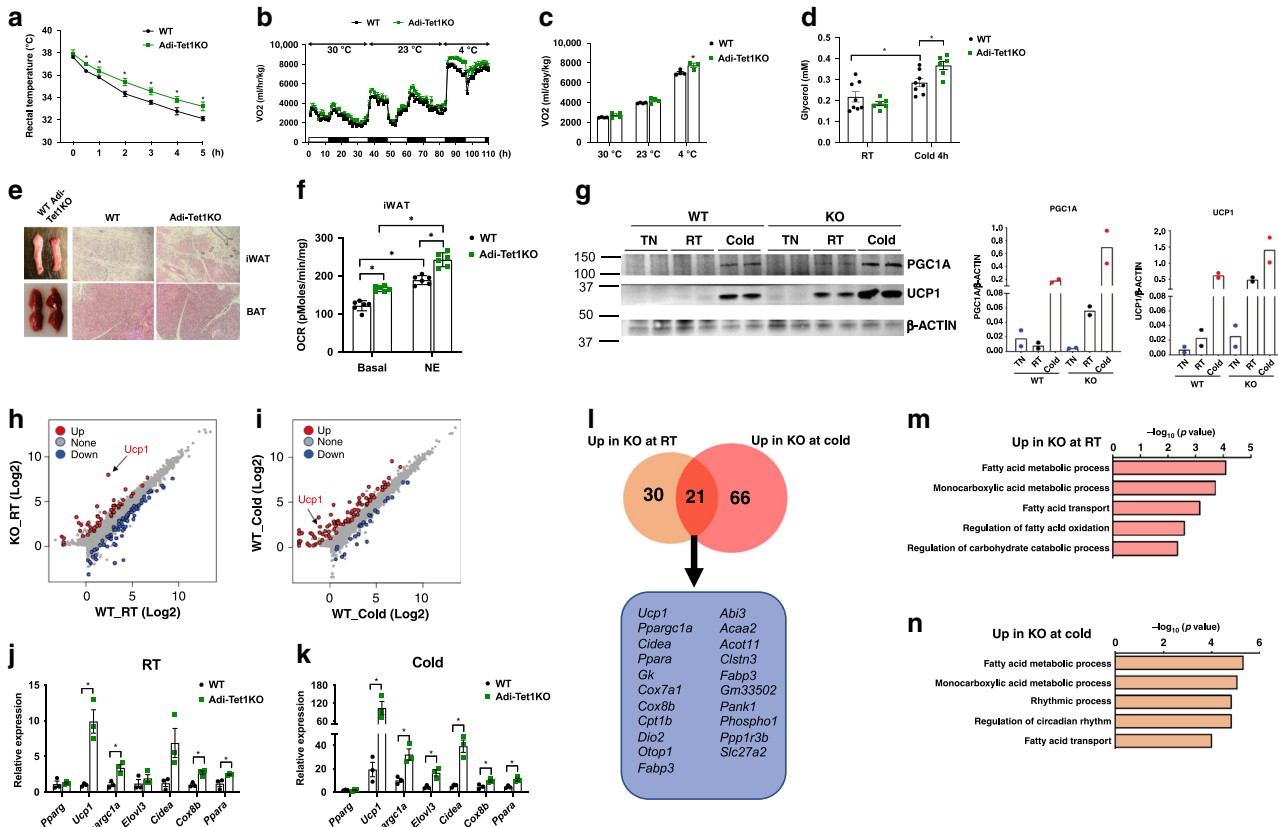

**Fig. 4 Adipose-specific *Tet1*-KO mice show improved cold tolerance. a** Rectal core body temperatures of Adi-Tet1-KO and WT mice under cold conditions at indicated time points ($n = 5$ per group. Data are expressed as means ± SEM. *denotes $p < 0.05$, determined by two-tailed Student's $t$ test and one-way ANOVA). **b**, **c** Shown is the whole-body oxygen consumption rate (VO2, **b**) and averaged VO2 **c** ($n = 4$ per group. Data are expressed as means ± SEM. *denotes $p < 0.05$, determined by two-tailed Student's $t$ test and two-way ANOVA followed by Bonferroni post-hoc testing). **d** Serum levels of glycerol were measured from WT and KO mice that were housed at RT or exposed to cold for 24 h ($n = 8$ WT, $n = 6$ KO. Data are expressed as means ± SEM. *denotes $p < 0.05$, determined by two-tailed Student's $t$ test and two-way ANOVA followed by Bonferroni post-hoc testing). **e** Whole tissue and H&E staining of iWAT and BAT from WT and KO mice that were exposed to cold for 7 days. **f** Oxygen consumption rate of iWAT from WT and KO mice was measured with and without stimulation of NE. Average basal respiration rate is presented with or without norepinephrine addition ($n = 6$ per group. Data are expressed as means ± SEM. *denotes $p < 0.05$, determined by two-tailed Student's $t$ test and two-way ANOVA followed by Bonferroni post-hoc testing). **g** Immunoblot of UCP1 and PGC1A from WT and KO iWAT from RT, exposed to TN or cold for 7 days. The relative expression is shown by normalizing to β-ACTIN ($n = 2$ per group. Data are presented as means of the two). **h**, **i** Scatter plot showing differentially regulated genes in inguinal KO adipocytes from WT and KO mice held at RT and exposed to cold for 4 h ($n = 2$ per group for RT, $n = 3$ WT, $n = 2$ KO for Cold, FC > 1.5, FDR < 0.05). **j**, **k** q-PCR analysis of key thermogenic genes from WT and KO inguinal adipocytes from WT and KO mice at RT and cold exposed for 4 h ($n = 3$ per group. Data are expressed as means ± SEM. *denotes $p < 0.05$, determined by two-tailed Student's $t$ test). **l** Venn diagram of the upregulated genes in the KO adipocytes at RT vs cold conditions and the gene names that are commonly upregulated in the KO adipocytes at both temperatures ($n = 2$ per group for RT, $n = 3$ WT, $n = 2$ KO for Cold). Gene list is provided in Supplementary Table 1. **m**, **n** Commonly upregulated biological pathways in the KO adipocytes from RT and cold. Source data are provided as a source data file.

at RT, and 87 genes were upregulated and 26 genes down-regulated after a 4-hr cold exposure (Fig. 4 h, i, l, Supplementary Fig. 9b). This suggests that TET1 acts as both a gene repressor and activator in inguinal adipocytes. Interestingly, the upregulated genes in the KO adipocytes greatly overlapped between RT and cold conditions (Fig. 4l), and many of those are involved in thermogenesis (e.g., *Ucp1*, *Pgc1a*, *Cidea*, and *Elovl3*). These were validated by qPCR analysis (Fig. 4 j, k). This suggests that the transcriptional regulation of key thermogenic genes in the KO is already primed under RT, which is mildly cold for mice. By contrast, little overlap was found between the genes that are downregulated in the KO adipocytes between RT and cold (Supplementary Fig. 9b). Gene Ontology term analysis found that the commonly upregulated pathways in KO inguinal adipocytes are relevant to thermogenesis, such as fatty acid metabolism, under both RT and cold (Fig. 4m, n).

**Tet1 KO mice are protected against diet-induced obesity.** Increased browning leads to increased energy expenditure and improved glucose homeostasis[18,19]. Thus, we assessed whether the increased browning during adipose-selective *Tet1* deficiency leads to metabolic improvement. Despite the body weight of Adi-Tet1KO remaining unchanged (Fig. 5a), their fat mass was significantly reduced compared to their littermate controls on a chow diet (Fig. 5b). Reduced adiposity in Adi-Tet1KO mice on chow was accompanied by improved glucose tolerance, insulin sensitivity, and hypoinsulinemia (Fig. 5c–e). Next, we asked whether adipose-selective *Tet1* deficiency confers protection against diet-induced obesity and impaired glucose tolerance by placing cohort mice on a high-fat diet (HFD, 60% calories from fat). The body weights of the two groups began to significantly diverge after 6 weeks of HFD feeding (Fig. 5f). The KO had reduced fat mass (Fig. 5g), and their tissue mass of inguinal and

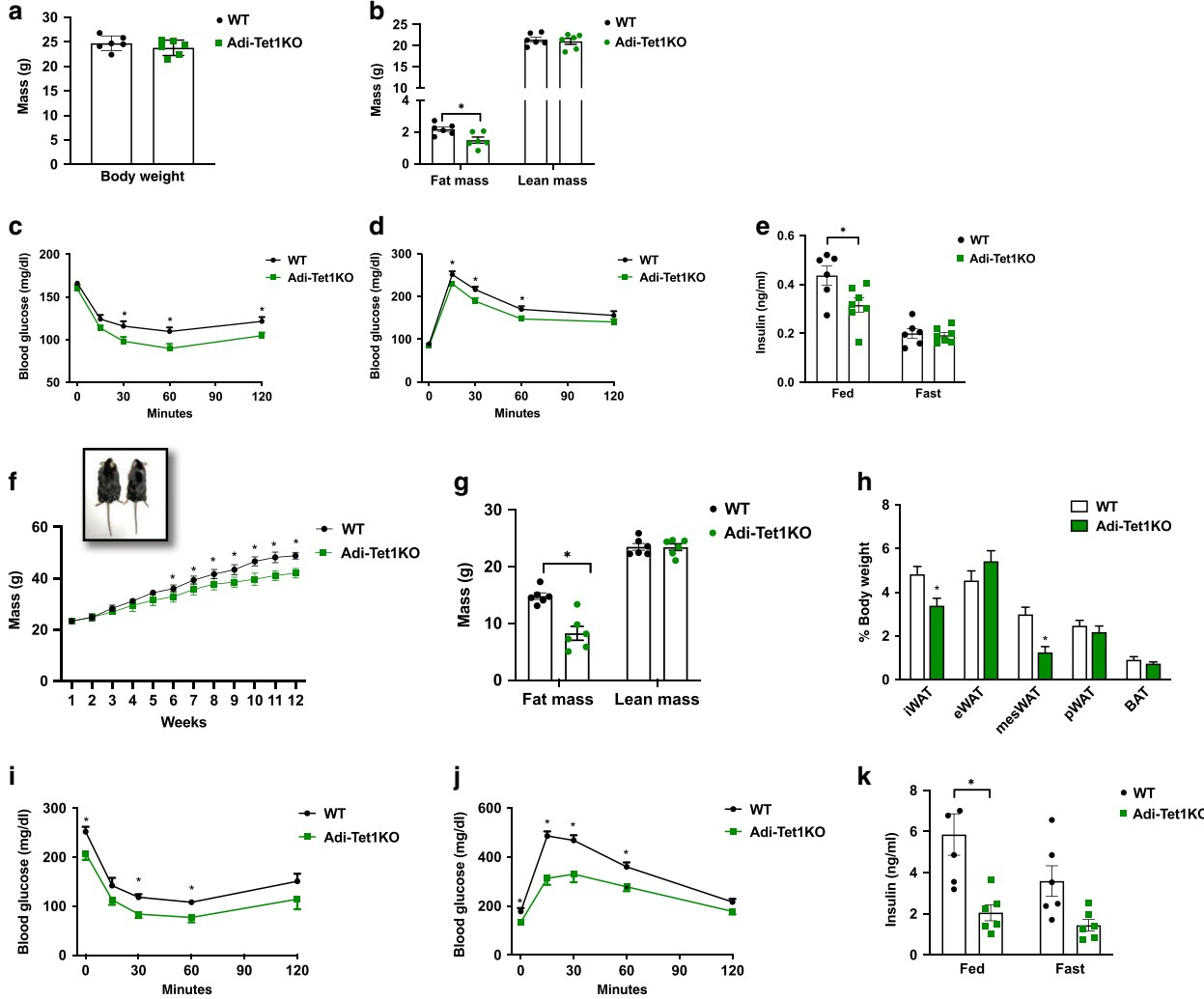

**Fig. 5 Adipose-specific *Tet1*-KO mice are protected from diet-induced obesity and metabolic dysregulation. a, b** Body weight (**a**) and body composition by EchoMRI (**b**) of 8-week-old WT and KO mice on a chow diet ($n = 6$ per group. Data are expressed as means ± SEM. *denotes $p < 0.05$, determined by two-tailed Student's *t* test). **c, d** Insulin tolerance test (**c**) and glucose tolerance test (**d**) on chow diet ($n = 8$ WT, $n = 6$ KO. Data are expressed as means ± SEM. *denotes $p < 0.05$, determined by two-tailed Student's *t* test). **e** Fed and fasted insulin levels from a chow-fed cohort ($n = 6$ per group. Data are expressed as means ± SEM. *denotes $p < 0.05$, determined by two-tailed Student's *t* test and two-way ANOVA followed by Bonferroni post-hoc testing). **f** Weekly body weight of WT and KO mice on a high-fat diet ($n = 7$ per group. Data are expressed as means ± SEM. *denotes $p < 0.05$, determined by two-tailed Student's *t* test). **g** Body composition after 10 weeks of HFD ($n = 6$ per group, Data are expressed as means ± SEM. *denotes $p < 0.05$, determined by two-tailed Student's *t* test). **h** Adipose tissue weight from HFD cohort ($n = 6$ per group. Data are expressed as means ± SEM. *denotes $p < 0.05$, determined by two-tailed Student's *t* test). **i, j** Insulin tolerance test (**i**) and glucose tolerance test (**j**) after 10 or 11 weeks on HFD, respectively ($n = 6$ per group, Data are expressed as means ± SEM. *denotes $p < 0.05$, determined by two-tailed Student's *t* test). **k** Fed and fasted insulin levels from high-fat-fed mice ($n = 6$ per group. Data are expressed as means ± SEM. *denotes $p < 0.05$, determined by two-tailed Student's *t* test and Two-way ANOVA followed by Bonferroni post-hoc testing). Source data are provided as a source data file.

mesenteric WAT was significantly decreased (Fig. 5h). Indirect calorimetry analysis by CLAMS revealed that the KO mice had increased energy expenditure (Supplementary Fig. 10a, b) without changes in food intake and locomotor activity (Supplementary Fig. 10c, d). Interestingly, the KO had an increased RER on HFD (Supplementary Fig. 10e), suggesting that they prefer carbohydrates over fat as a main energy source on HFD. Consistent with the lean phenotype, KO mice were more glucose tolerant and insulin sensitive than controls, having reduced insulin levels at fed and fast states on HFD (Fig. 5i–k).

**TET1 acts in a DNA demethylase-independent manner.** TETs mediate their biological functions in both DNA demethylase-dependent and -independent manners[27–31]. Thus, we addressed whether demethylation activity is necessary for TET1 to suppress

the thermogenic program in beige adipocytes. First, as a genetic approach, we generated various *Tet1* mutant alleles (Fig. 6a). Lentiviral overexpression of a *Tet1* mutant allele that lacks c-terminal catalytic activity (Tet1ΔCD)[42] (Fig. 6b) still repressed *Ucp1* transcription to a similar degree as the wild-type allele (Tet1 WT) (Fig. 6c). On the other hand, overexpressing either the truncation mutant that contains only the catalytic domain (Tet1 CD) or the catalytically inactive mutant with two critical amino acid substitutions (Tet1 CDM)[43] was not able to repress *Ucp1* expression (Fig. 6c). Both Tet1 WT and Tet1 ΔCD equally inhibited mitochondrial respiration in the presence of nor-epinephrine (Fig. 6d, e).

Second, we performed *Tet1* loss-of-function studies in cells that had all three DNA methyltransferases (*Dnmts*) knocked down to reduce the level of 5mC (Supplementary Fig. 11a), which is the

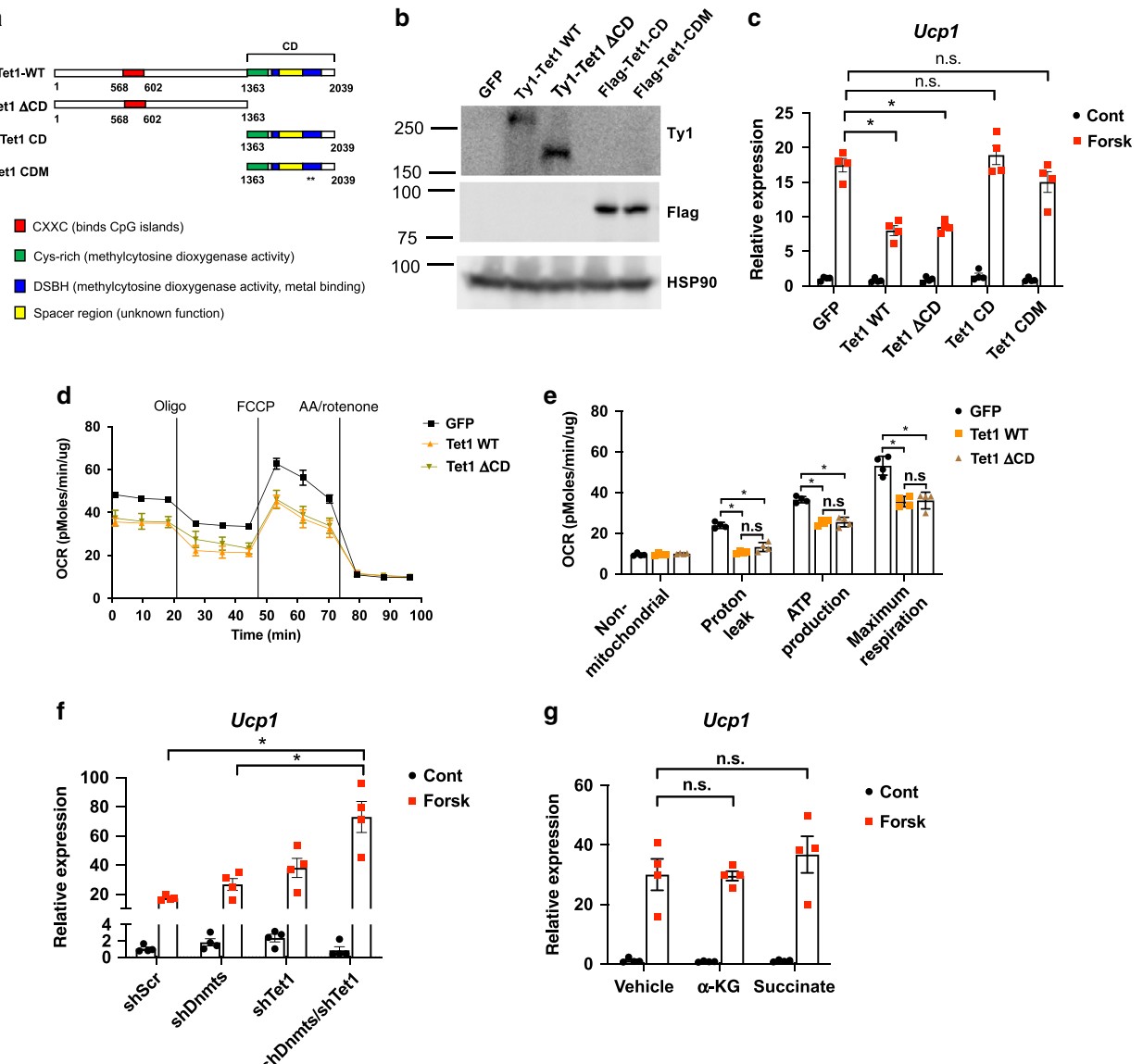

**Fig. 6 TET1 suppresses the thermogenic gene program in a DNA demethylase–independent manner. a** The protein map showing *Tet1* wild-type and mutant alleles: the *Tet1* wild-type (Tet1 WT) allele, a truncation mutant that lacks DNA demethylase activity (Tet1 ΔCD), a mutant that contains only DNA demethylase activity (Tet1 CD), and a catalytically inactive form (Tet1 CDM). **b** Mature beige adipocytes were transduced with lentiviral expression plasmids for the various *Tet1* alleles from (**a**). **c** The basal- and forskolin-stimulated levels of *Ucp1* were measured by qPCR from (**b**) (*n* = 4 per group. Data are expressed as means ± SEM. *denotes *p* < 0.05, determined by two-tailed Student's *t* test and one-way ANOVA followed by Bonferroni post-hoc testing). **d, e** Mitochondrial respiration is measured by Seahorse from beige adipocytes that overexpress Tet1 WT, Tet1 ΔCD, or GFP. (*n* = 4 per group. Data are expressed as means ± SEM. *denotes *p* < 0.05, determined by two-tailed Student's *t* test and two-way ANOVA followed by Bonferroni post-hoc testing). **f** Beige adipocytes were transduced with hairpins against *Dnmt1, 3a*, and, *3b* or control scramble RNA (shScr), and differentiated beige adipocytes were transduced with shTet1. The basal- and forskolin-stimulated levels of *Ucp*1 were measured by qPCR (*n* = 4 per group. Data are expressed as means ± SEM. *denotes *p* < 0.05, determined by two-tailed Student's *t* test and two-way ANOVA followed by Bonferroni post-hoc testing). **g** Alpha-KG (α-KG), diethyl-succinate (Succinate), or a vehicle was used to pre-treat mature beige adipocytes for 3 hr. The basal- and forskolin-stimulated levels of *Ucp1* were measured by qPCR (*n* = 4 per group. Data are expressed as means ± SEM. *denotes *p* < 0.05, determined by two-tailed Student's *t* test and two-way ANOVA followed by Bonferroni post-hoc testing). Source data are provided as a source data file.

enzymatic substrate for TET. If DNA demethylation activity was critical, the effect of *Tet1* loss-of-function would be diminished in these cells. Interestingly, knocking down the *Dnmts* alone resulted a trend toward an increase in the *Ucp1* expression, and double knockdown with *Tet1* further increased the expression of *Ucp1* (Fig. 6f). Third, we manipulated TET demethylase activity at the co-factor level. Alpha-ketoglutarate (α-KG), a key metabolite from the TCA cycle, acts as a co-factor for the demethylation activity of TETs, whereas another TCA intermediate, succinate,

inhibits TET activity[44,45]. We treated mature brown and beige adipocytes with cell-permeable forms of α-KG and diethyl-succinate then subjected them to forskolin treatment. Neither metabolite had any significant impact on forskolin-stimulated *Ucp1* expression (Fig. 6g).

Third, we profiled genome-wide maps of 5hmC, a major intermediate product of TET-mediated oxidation of methylated cytosine (5mC), using the sensitive 5hmC-Seal-Seq method in iWAT from C57BL/6 J WT mice housed at RT or exposed to cold

for 7 days (Supplementary Fig. 12). We found no significant changes in 5hmC levels at the key thermogenic genes under cold conditions, as exemplified at the *Ucp1* and *Ppargc1a* loci (Supplementary Fig. 12a, b). This suggests that 5hmC accumulation is not a key epigenetic event underlying the temperature-triggered browning of beige adipocytes. Taken together, these results suggest that TET1-mediated suppression of the thermogenic gene program is largely done in a DNA demethylase-independent manner.

Lastly, we performed bisulfite PCR sequencing to assess the base-pair resolution of DNA methylation changes at several CpG-rich *Ucp1* and *Tet1* promoter/enhancer regions[46–49]. We noted an overall reduction of CpG methylation at one of the *Ucp1* promoter regions (named *Ucp1* (#1) under lower temperature conditions (Supplementary Fig. 13a, e). Other than that, no obvious changes in average CpG methylation were detected in the *Ucp1* and *Tet1* promoter/enhancer regions that we tested (Supplementary Fig. 13b–d, f–h, o–q, r–t). In terms of methylation at individual CpGs, some of the CpG sites tended to be hypermethylated while some were rather hypomethylated at lower temperatures or showed no particular correlation with temperature conditions (Supplementary Fig. 13i–l, u–w). Future studies are warranted to determine the functional significance of the changes at individual CpG sites.

**TET1 coordinates with HDAC1 to mediate the thermogenic gene repression**. It has been proposed that the repressor role of TET1 in gene regulation in other cell types is accomplished through interacting with other repressor proteins like polycomb repressive complex 2 (PRC2)[50], HDACs[51], and SIN3A[52]. Since genetic and pharmacological inhibition of HDAC1 increases UCP1 and PGC1α expression and oxidative metabolism[53–55]. we sought to determine whether TET1-mediated repression involves interacting with HDAC1. First, we detected an interaction between TET1 and HDAC1 by co-immunoprecipitating in HEK293T cells (Fig. 7a). Interestingly, the interaction with HDAC1 still existed with Tet1 ΔCD, which lacks DNA demethylase activity at the c-terminal region (Fig. 7a). We examined whether TET1 and HDAC1 are recruited to the same gene regulatory regions of *Ucp1* and *Ppargc1a*[55]. Since we were not able to identify a high-quality antibody against endogenous TET1, we immunoprecipitated TET1 tagged with of Ty1 tag from overexpressor cells. With ChIP-reChIP analysis, we confirmed the simultaneous binding of HDAC1 and TET1 at these regulatory regions (Fig. 7b) and that both TET1 and HDAC1 binding at these sites was greatly diminished with forskolin stimulation (Fig. 7c–e).

HDAC1 deacetylates histones, which prevents transcription. Therefore, to examine the downstream epigenetic changes that TET1 loss-of-function confers, we performed ChIP-PCR using the active histone mark H3K27ac in WT and KO adipose tissue harvested from different temperature conditions. As expected, in WT iWAT samples, H3K27ac enhancer activity was higher at RT and highest during cold exposure as compared to TN. Overall, this temperature-dependent H3K27ac enhancer activity was even more dramatic in KO iWAT (Fig. 7f–h), in concert with increased expression of *Ucp1* and *Ppargc1a* (Fig. 4j, k). Lastly, we assayed the role of HDAC1 in TET1-mediated thermogenic gene suppression. CRISPR-Cas-mediated knockout of HDAC1 (Supplementary Fig. 11b) fully rescued TET1-mediated repression of *Ucp1* (Fig. 7i, j). Together, our results suggest that the role of TET1 as a suppressor of the thermogenic gene program is in large part due to HDAC1.

## Discussion

While classical brown and beige adipocytes share many fundamental features in morphology and function, discrete characteristics have been identified, including the plasticity of beige thermogenesis[1]. A recent epigenomic profiling study demonstrated that the temperature-mediated plasticity of sub-cutaneous fat is accompanied by profound changes in chromatin state[10]. To add to these results, we identified TET1 as a beige fat-selective epigenetic repressor of the thermogenic gene program.

We demonstrated that TET1 represses thermogenic gene regulation in a largely DNA demethylase-independent manner. In support of this notion, transcriptional changes induced by over-expression of TET1 were highly similar to those induced by its demethylation activity-dead mutant in differentiated cell lines. Other studies have demonstrated that TETs, in addition to modifying cytosine methylation, can act as gene regulators through a DNA methylation-independent manner[28]. The repressive role of TET1 in transcriptional regulation has been proposed to derive from its interaction with other repressor proteins. In ES cells, but not in somatic cells, TET1 contributes to silencing some genes by interacting with polycomb repressive complex 2 (PRC2), which targets repressive histone mark H3K27me3[50]. TET1 is also found in the repressor complex containing SIN3A and HDAC1/2 in both mESCs and HEK293T cells, and its localization greatly overlaps with the SIN3A binding profile genome-wide[30]. Thus, our results are in line with these studies, as TET1 cooperates with HDAC1 to repress some of the key thermogenic genes like *Ucp1* and *Ppargc1a*. We speculate that such interaction facilitates local epigenetic modifications and regulates other transcription factors and co-factors to regulate transcriptional activity.

There is also indirect evidence that DNA demethylation and TETs are also required for the development of interscapular BAT[56]. In this study, the authors suggested that TET proteins mediate demethylation at the promoter region of *Prdm16*[56], a key developmental gene that is critical for the brown adipogenic lineage and maintains brown adipocyte identity. Moreover, reducing the level of α-ketoglutarate (α-KG), a co-factor for the TET enzymes, leads to reduced demethylation of *Prdm16* and impaired brown adipocyte development and function in mutant mice carrying loss-of-function AMPKa1[56]. Given the pro-adipogenic function of TET1 in white and brown adipogenesis, our results demonstrating TET1 is an inhibitor of thermogenic genes may seem contradictory. However, such dual roles have been reported with other "whitening" transcription factors. For example, Zfp423, preferentially expressed in white adipocytes, is a pro-adipogenic commitment factor in vitro and in vivo[57], but it also acts as a molecular gate keeper that maintains white cell identity while suppressing browning in mature adipocytes[58]. Interestingly, a recent study reported that the glucocorticoid receptor (GR) drives beige adipocyte whitening as an upstream regulator of Zfp423[10]. TLE3, another whitening transcription factor, promotes adipose conversion during early differentiation by interacting with PPARγ and antagonizing Wnt signaling[59]. But a recent study identified its novel role as a suppressor of thermogenic gene expression in beige adipocytes[60]. Notably, both Zfp423 and TLE3 inhibit the activity of pro-browning transcription factor early B-cell factor 2 (EBF2)[58,60], which cooperates with PPARγ and epigenetic modifiers, such as the chromatin remodelers BRG1 and BAF, and a long noncoding RNA, Blnc1[61]. Future studies are warranted to determine whether TET1-mediated gene repression converges with these known transcriptional regulators.

Notably, two global studies suggest that differences in the DNA methylation profile is not a major contributor to cell type-specific gene expression in white vs beige or brown adipocytes[62,63]. The first global study employed restriction landmark genomic scanning (RLGS)[63], the method by which methylation-sensitive restriction enzymes preferentially cut CpG islands in regulatory regions[63]. In this study, authors also did not find a dramatic

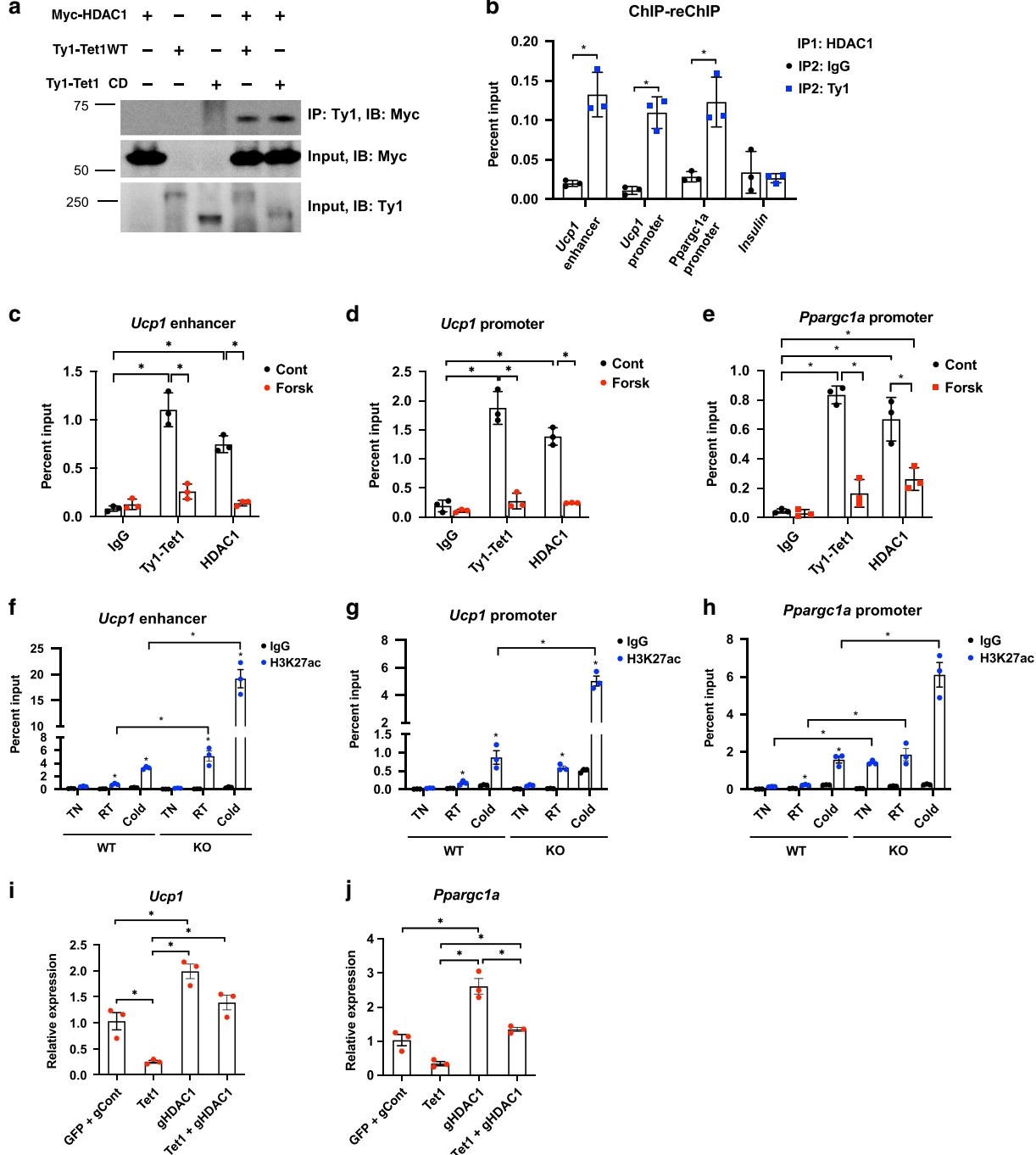

**Fig. 7 TET1 coordinates the epigenetic remodeling of the regulatory regions of *Ucp1* and *Ppargc1a* by coordinating with HDAC1. a** Co-immunoprecipitation assay was performed using protein lysates from HEK293T cells that were co-transfected with vectors expressing Ty1-Tet1 WT, Ty1-Tet1deltaCD, and Myc-HDAC1. **b** ChIP-reChIP qPCR analysis was performed on beige adipocytes that were transduced with Ty1-Tet1 WT or GFP using HDAC1 (1st IP), Ty1 (2nd IP), or IgG as a control. The enrichment efficiency is presented as a percent input at the indicated binding sites ($n = 3$ per group. Data are expressed as means ± SEM. *denotes $p < 0.05$, determined by two-tailed Student's *t* test followed by Bonferroni post-hoc testing). **c–e** Ty1 and HDAC1 ChIP-qPCR analysis was performed in cells that express Ty1-Tet1 WT with and without forskolin stimulation ($n = 3$ per group. Data are expressed as means ± SEM. *denotes $p < 0.05$, determined by two-tailed Student's *t* test and two-way ANOVA followed by Bonferroni post-hoc testing). **f–h** H3K27ac ChIP-qPCR analysis was performed in WT and KO iWAT from RT or exposed to TN or cold for 7 days ($n = 3$ per group). Data are expressed as means ± SEM. *denotes $p < 0.05$, determined by two-tailed Student's *t* test and two-way ANOVA followed by Bonferroni post-hoc testing. **i, j** Beige adipocytes were transduced with either *Tet1* overexpressor, gRNA against *Hdac1*, both, or control plasmids. The forskolin-stimulated level of *Ucp1* and *Ppargc1a* was measured by qPCR ($n = 4$ per group, Data are expressed as means ± SEM. *denotes $p < 0.05$, determined by two-tailed Student's *t* test and two-way ANOVA followed by Bonferroni post-hoc testing). Source data are provided as a source data file.

difference in the DNA methylation profile between primary white vs. brown adipocytes. The other global study profiled in vitro-differentiated white and brown adipocytes from inguinal and brown adipose depots using reduced representation bisulfite sequencing (RRBS)[62,63], which focuses on CpG-rich promoter methylation. Although an overall negative correlation between promoter methylation and gene expression was observed when comparing white and brown adipocytes, dramatic differences in DNA methylation at the thermogenic genes were not reported. Consistent with these reports, we did not detect dramatic changes in DNA demethylation at the key thermogenic genes. However, we observed that some of the CpG sites displayed simultaneous hyper- or hypo-methylation depending on temperature changes at the *Ucp1* promoter region. Thus, genome-wide methylation profiling studies are warranted to more accurately understand the DNA methylation profile under different temperature conditions. Moreover, functional validation studies will be necessary at the level of individual CpGs.

One of the major caveats of our study is the use of Fabp4-Cre, which is expressed in several other non-adipose tissues[35,64]. Thus, we conducted additional experiments to assess whether there is an additional contribution from non-adipose tissues to the increased thermogenesis in Adi-Tet1 KO mice. First, our in vitro gain- and loss-of-function studies showed that *Tet1* knockdown and over-expression have a beige adipocyte-autonomous effect on thermogenic gene regulation and mitochondrial respiration. Second, we showed that inguinal-selective knockdown of *Tet1* by AAV8-hAdi-Cre and noted a similar effect on thermogenesis. Third, to address if there is input from macrophage cells in Adi-Tet1 KO mice, we conducted co-culture experiments using beige adipocytes and primary macrophages isolated from Adi-Tet1 KO vs. WT mice. There were no significant changes in basal and forskolin-induced thermogenic gene expression. Lastly, we report no major phenotypic changes in thermogenesis and energy homeostasis in the PDGFRa-Tet1 KO generated using PDGFRa-Cre, which is expressed in preadipocytes and a variety of tissues, including the lung, heart, intestine, skin, and cranial facial mesenchyme[37–39,65–67]. We, however, note that it is still possible that there could be an additional contribution on top of that from adipose tissues.

In summary, we identified TET1 as an important epigenetic regulator of the thermogenic gene program in beige adipocytes that coordinates with HDAC1. Adipose-specific TET1 loss-of-function led to increased energy expenditure and protection from diet-induced obesity, insulin resistance, and glucose tolerance.

## Methods

**Cell culture**. Immortalized beige and brown adipocytes were maintained and differentiated with an adipogenic cocktail (0.5 mM IBMX, 5uM Rosiglitazone, 5 mg/ml insulin, 1 mM dexamethasone). To generate lentivirus particles, lentiviral constructs were co-transfected with pM2DG- and psPAX-expressing plasmids into 293 T cells. After 48 h, virus-containing supernatant was collected, filtered through 0.45 μm filters, and added to mature day 5 adipocytes for 48 h along with 8 μg/ml Polybrene. Transduction efficiency was determined by comparing to cells transduced in parallel with a GFP-expressing lentivirus. For the ex vivo system, subcutaneous WAT and iBAT from wild-type C57BL/6 mice was fractionated with digestion buffer (10 mg/ml collagenase D, 2.4 units of dispase II, 10 mM CaCl₂ in PBS). The stromal-vascular fraction (SVF) were isolated by 1.5 u/ml collagenase and plated in culture and differentiated with adipogenic cocktail. For co-culture experiment, beige adipocytes were cultured in 6-well plates and differentiated at day 8, and macrophages (2 ×10⁵ cells/well) were plated onto the transwell insert containing a 0.4 μm polyethylene terephthalate membrane (Costar, Corning, USA) in serum free medium. After incubation together for 24 h, the transwell was removed and beige adipocytes were harvested for analysis. 3T3-L1 adipocytes were differentiated in 12-well plates and then treated with conditioned medium from LPS-activated macrophages from WT and KO mice. 3T3-L1 adipocytes were cultured for 24 h before gene expression analysis.

**Reagents**. Insulin, dexamethasone, isobutylmethylxanthine (IBMX), α-KG, CL316, 243, diethyl succinate, and thyroid hormone (T3) were purchased from Sigma. Rosiglitazone was purchased from Cayman. Antibodies were purchased from

GeneTex (Tet1, GTX1242071), Thermo Fisher (β-actin, MA5-14739), Cell Signaling (HSP-90, 4874), GenScript (Ty1, A01004), Covance (HA, MMS-101R), Sigma (Flag, F3165), Santa Cruz (Myc, SC-40), Abcam (UCP1, Ab10983; PGC1α: Ab54481; H3K27ac: Ab4729), Biolegend (CD45-PerCP/Cy5.5,103131; F4/80-PE/Cy7, Cat# 123113; Cd11b-Pacific Blue, 101223; CD301-APC, 145707 and eBioscience (Cd11c-PE,12-0114-81).

**Animals**. Tet1^f/f mice were obtained from Dr. Anjana Rao laboratory at UCSD. Mice were maintained under a 12 h light/12 h dark cycle at room temperature (23 °C) with free access to food and water. For thermal challenge, mice were placed in a cold chamber (4 °C) for up to 1 week or at thermoneutrality (30 °C). Body temperature was measured using a rectal probe (Physitemp). For high-fat feeding studies, male C57BL/6J mice were put on the diet beginning at 8 weeks of age and continuing for up to three months. Blood and various tissue samples were collected. For the CL316,243 studies, animals were given the drug by IP injection for 5 consecutive days at RT and euthanized for the gene expression analysis. For histology, adipose tissues were fixed with neutral-buffered formalin and embedded in paraffin, and sections were stained with H&E.

iWAT-specific *Tet1* Adi-CRE mice were generated by injecting adeno-associated virus (AAV) expressing Adiponectin CRE (AAV8-hAdp-iCre, Vector Biolabs) or control Null (AAV8-Null) into inguinal WAT level. Briefly, mice were anesthetized and maintained using isoflurane. Area around iWAT was shaved on either side of the mice and cleaned with Betadine (#19-027132,) and Ethanol. A small incision was made to expose the iWAT pads. AAV was injected into both the iWAT depot at 10 different locations per depot of *Tet1*f/f adult mice using hamilton microsyringe. Viral titer of 5.0 × 10¹¹ genomic copies (GC) per mouse. After injection, the incision was closed using suture (#101-7137, Henry Schein). Efficacy of viral infection and knockdown was evaluated by quantification of TET1 expression. All animal work was approved by the UC Berkeley ACUC.

**Protein analysis**. Whole-cell protein lysates were prepared using RIPA lysis buffer (1% Triton X-100, 1% Sodium deoxycholate, 0.1% SDS, 0.15 M NaCl, 50 mM Tris (pH 7.2) and protease inhibitor cocktail (Complete Mini-EDTA free, 11836170001, Roche). Protein was resolved using Tris-glycine gels and transferred to PVDF membrane. After blocking with 5% nonfat dried milk in TBS-Tween (0.25%), the membranes were incubated with the appropriate primary antibodies and loading control. Immunoblots were quantified by the ImageJ program.

**Immunoprecipitation**. HEK-293T cells were transfected with various DNA constructs using Lipofectamine 3000 (Invitrogen). A day after transfection, cells were lysed with RIPA buffer with protease inhibitor cocktail. 500 mg of protein was incubated with the appropriate antibodies overnight. The next day, protein A/G PLUS-Agarose (SC-2003, Santa Cruz) was added and incubated for 2 h, washed with lysis buffer five times and PBS once. Beads were eluted with non-reducing SDS/PAGE loading buffer and subjected to SDS/PAGE and western blotting.

**Dot-blot analysis**. Genomic DNA was extracted using the DNeasy Blood & Tissue kit (Qiagen, 69504) following the manufacturer's protocol. DNA was denatured at 95 °C for 15 min in 0.1 M NaOH. It was neutralized with 1 M NH4OAc on ice and diluted to 300 mg in DNAse free water. 200 ng DNA was applied to a positively charged nylon membrane under vacuum with a Dot Blot Microfiltration Apparatus (Bio-Rad). The membranes were briefly washed in 2× SSC buffer (0.3 M NaCl, 30 mM sodium citrate) for 5 min, then baked at 80 °C for 5 min, and then cross-linked using a UV Stratalinker 1800. Membranes were blocked with 5% nonfat dried milk in TBS-Tween (0.25%). The membranes were incubated with the 5mC primary antibodies. For a loading control, membranes were stained with methylene blue. Immunoblots were quantified by the ImageJ program.

**Indirect calorimetry**. Metabolic rate was measured by indirect calorimetry in open-circuit Oxymaxchambers, a component of the Comprehensive Lab Animal Monitoring System (CLAMS; Columbus Instruments). Mice were housed individually at various temperatures under a 12 h light/12 h dark cycle. Food and water were available ad libitum.

**Cellular and tissue respiration**. For cellular respiration, lentivirally transduced beige adipocytes were plated on XF24 Cell Culture Microplates. For tissue respiration, freshly isolated iWAT and iBAT were rinsed in sterile saline and dissected with a microdisector to 2 mm per well. Oxygen consumption rate (OCR) was determined using an XF24 Extracellular Flux Analyzer (Seahorse Bioscience). Uncoupled and maximal OCR was determined using oligomycin (4 μM) and FCCP (4 μM). Antimycin A and rotenone (2 μM each) were used to inhibit Complex III- and Complex I-dependent respiration.

**ChIP-qPCR**. Cells were cross-linked with 1% formaldehyde for 10 min at room temperature. Cross-linked chromatin was sonicated using an S220 Ultrasonicator (Covaris) to generate DNA fragments of ∼200–500 bp. Inputs were taken from cleared lysates, and the rest were rotated O/N at 4 °C with Ty1, HDAC1, and IgG antibodies for immunoprecipitation. An aliquot of 20 μl of pre-washed Dynabeads

Protein G was added per IP and rotated 1 h at 4 °C. Beads were successively washed in low-salt RIPA buffer (20 mM Tris-HCl [pH 8.0], 1 mM EDTA, 1% Triton x-100, 0.1% SDS, 140 mM NaCl, 0.1% Na deoxycholate), high-salt RIPA buffer (20 mM Tris-HCl [pH 8.0], 1 mM EDTA, 1% Triton x-100, 0.1% SDS, 500 mM NaCl, 0.1% Na deoxycholate), LiCl buffer (250 mM LiCl, 0.5% NP40, 0.5% Na deoxycholate, 1 mM EDTA, 10 mM Tris-HCl [pH 8.0]) and TE buffer (10 mM Tris-HCl [pH 8.0] and 1 mM EDTA). Each reaction was then incubated in digestion buffer (50 mM Tris-HCl [pH 8.0], 1 mM EDTA, 100 mM NaCl, 0.5% SDS, proteinase K) for a minimum of 4 h at 65 °C to reverse cross-links. DNA was recovered using a phenol-chloroform extraction. ChIP-reChIP was performed in essentially the same way as ChIP, except that the first elution was carried out in a digestion buffer (50 mM Tris, pH 8.0, 1 mM EDTA, 1% SDS, 50 mM NaHCO₃) at 65 °C for 10 min. After saving the supernatant, the beads were with 40 ul 10 mM DTT for 30 min at 37 °C. The combined elutes were subjected to the second IP. Real-time qPCR primers are listed in Supplementary Table 2. All data were normalized to input.

**RNA extraction and quantitative PCR.** Total RNA was extracted from cells or tissues using TRIzol reagent according to the manufacturer's instructions. cDNA was reverse-transcribed from 1 μg of RNA using the RETROscript first strand synthesis kit (Ambion). Quantitative PCR (qPCR) was performed with SYBR Green qPCR Master Mix (Applied Biosystems) using a 7900HT Fast Real-Time PCR System (Applied Biosystems) and CFX96 Touch (BioRad). Primer sequences are listed in Supplemental Table 2. The relative amount of mRNA normalized to cyclophilin B was calculated using the delta–delta method[68].

**RNA-Seq analysis.** RNA samples were extracted using the RNeasy Mini kit (Qiagen, 74104) following the manufacturer's protocol. Libraries were prepared using the BGI Library Preparation Kit, and sequencing was performed on the BGISEQ. RNA-Seq reads were aligned to UCSC mm10 genome using STAR aligner[69] with an option, "--outFilterMultimapNmax 1". Mitochondrial reads were filtered out to avoid sequencing depth bias due to mitochondrial abundance. Then, raw read count for each gene was measured using Feature Counts. Differential gene expression analysis was performed using edgeR. Hierarchical clustering was performed using group-wise average gene expression levels to identify distinct functional modules of genes using Ward's criterion and Pearson correlation as a similarity measure. Gene ontology analysis was done using EnrichR.

**5hmC-seal analysis.** Genomic DNA (250 ng) was sonicated to ~100–500 bp with a Bioruptor PICO sonicator (Diagenode). Sonicated DNA was end-repaired, A-tailed, and ligated to paired-end adapters following the standard Illumina protocol. The glucosylation reactions were performed in a 50 μl solution containing 1x glucosylation buffer, above adapter-ligated DNA, 200 μM UDP-Azide-Glucose (Active Motif, 55020), and 5 U T4 ß-glucosyltransferase (Thermofisher, EO0831), at 37 °C for 1 h. After glucosylation, the reaction was purified by Zymo DNA clean & concentrator Kit (Zymo, D4014) and eluted into 45ul ddH2O. Then, 1.5 μl DBCO-PEG4-Biotin (Click Chemistry Tools, A105. 4.5 mM stored in DMSO, dilute from 30 mM stock before use) was added to the 45ul glusosylated DNA and the reactions were incubated at 37 °C for 2 h. Next, the DNA was purified by Zymo DNA clean & concentrator Kit and eluted in 10ul ddH2O. The purified DNA was pulled down by 5 μl streptavidin C1 beads (Thermofisher, 65001) for 15 min according to the manufacture's instruction. The beads were subsequently undergone ten washes with 1x binding-washing buffer and two washes with ddH2O and were resuspended in 15ul ddH2O. All binding and washing were done at room temperature. The captured DNA fragments were amplified with 12 cycles of PCR amplification using the Phusion DNA polymerase. The PCR products were purified using 1.0X AMPure XP beads according to the manufacture's instruction. DNA concentration of each library was measured with a Qubit fluorometer (Life Technologies) and sequencing was performed on the Next-Seq instrument (Illumina). 5hmC-seq analysis: Sequencing reads was trimmed adapter using cutadapt (https://cutadapt.readthedocs.io/en/stable/), aligned to mouse genome mm10 using BWA with default parameters http://bio-bwa.sourceforge.net/. MethylQA was used to process the aligned BAM files. Concordantly aligned read-pairs were selected and deduplicated using Picard tool (https://broadinstitute.github.io/picard/). Genome browser tracks were created in bigwig files using "makeUCSCfile" in Homer and bedGraphToBigWig in UCSC toolkit.

**Bisulfite sequencing.** Bisulfite modification was carried out with 250 ng genomic DNA using the EZ DNA Methylation-Lightning Kit (#D5030 Zymo Research, Irvine, CA) according to the manufacturer's instructions. Promoter and enhancer regions within the *Ucp1* and *Tet1* genes were amplified using bisulfite PCR. Bisulfite PCR primers were designed with the BiSearch and Methfinder programs, which are listed in Supplementary Table 2. The purified PCR products were cloned into the Topo TA Cloning System (Invitrogen, Carlsbad, CA, USA). Colonies were screened and a minimum of ten clones from each single bisulfite-treated DNA were sequenced, and the data were analyzed for statistical significance.

**FACS analysis.** SVF cells were incubated with indicated antibody for 20 min in dark, washed, span at 300*g* for 5 min, resuspended in FACS buffer (1× PBS containing 0.5% BSA) and passed through 40 μm filter prior to FACS analysis. FACS was performed on BD Influx cell sorter. Cells were initially chosen based on forward and side scatter (FFS and CCS) and trigger pulse width. Cells that were not incubated with antibody were used as a control to determine background fluorescence levels.

**Plasmids.** Hairpins against Dnmt1, Dnmt3a, and Dnmt3b are from Sigma. HDAC1 was subcloned to pcDNA3.1 at EcoRI/NotI for Myc-HDAC1. Lentiviral overexpression vectors for Tet1WT, Tet1ΔCD, Tet1CD, and Tet1CDM were subcloned into pCDH using various multicloning sites (XbaI/NotI for Ty1-Tet1WT, Ty1-Tet1ΔCD, XbaI/NheI for Flag-Tet1CD, Flag-Tet1CDM). Hairpins targeting *Tet1* were subcloned at AgeI/EcoRI or purchased from Open Biosystems. Hairpin sequences are shown in Supplemental Table 1. sgRNAs that targeted *Hcac1* were cloned into lentiCRISPR v2 vector. Hairpin and sgRNA sequences are shown in Supplementary Table 1.

**Statistical analyses.** Data are presented as means ± SEM and individual data points are plotted. Sample size was determined by our experience with inherent variability. No statistical method was used to predetermine sample size. Statistical analyses and the number of samples (*n*) were described in detail for each figure panel. Statistical analyses and the number of samples (*n*) is described in detail for each figure panel. Two-tailed unpaired Student's *t* test was used for the comparison between two groups. One-way analysis of variance (ANOVA) or two-way ANOVA followed by the Bonferroni's test was used for the multiple comparisons. Statistical analyses were performed using excel and GraphPad Prism. All reported *p* values were two-sided and differences were considered significant at *p* < 0.05.

**Reporting summary.** Further information on research design is available in the Nature Research Reporting Summary linked to this article.

## Data availability

The authors declare that the data supporting the findings of this study are available within the paper and its supplementary information files. Global profiling data are available in the GEO repository under accession number: GSE153093. Source data are provided with this paper.

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

## Acknowledgements

Work was funded by AHA Award # 19POST34380834 to D.Y. and R01 DK116008 to S.K. We thank Drs. Hei Sook Sul and Jen-Chywan Wally Wang (UC Berkeley) for helpful conversations about manuscript and thank Dr. Shingo Kajimura (UCSF) for allowing us to learn adenoviral injection to inguinal fat pad in his laboratory.

## Author contributions

S.K. supervised experiments and wrote the manuscript. Experiments were carried out by S.D.V., D.Y., J.K., H.X., P.A., and S.K. H.L. analyzed RNA-Seq and 5hmC-Seal analysis. P.H.J. helped with CLAMS studies. Y.O. helped with adenovirus injections.

## Competing interests

The authors declare no competing interests.
