## [Peer Review File · Nature Communications]

Peer Review File - Reviewers' comments first round:

Reviewer #1 (Remarks to the Author):

This is a well-written manuscript reporting that the DNA dioxygenase Tet1 is a thermosensitive repressor of the thermogenesis program in beige fat. Expression of Tet1 is suppressed by cold temperature, which then facilitates induction of key thermogenic genes. Using adipose-selective KO mice, the authors showed metabolic changes in KO mice that improved cold tolerance and increased energy expenditure. Interestingly, thermogenic genes were already up-regulated at RT in inguinal adipocytes of Tet1 KO mice, suggesting an epigenetic mechanism that enhances thermogenic gene expression during cold exposure when Tet1 is repressed. The authors further demonstrate that the gene suppressive role of Tet1 is DNA demethylase-independent and involves coordination with HDAC1. Overall, the study is interesting and proposes Tet1 as a potential therapeutic target to increase energy expenditure in obesity and related metabolic syndromes. However, there are some aspects of the study that need improvement.

Major comments:

1. The claim that TET1 is a powerful epigenetic regulator of the thermogenic gene program in beige adipocytes is over-stated. While the study shows an involvement of Tet1, there is no evidence that Tet1 is sufficient to repress thermogenesis in the absence of co-repressors such as HDAC1.
2. Figure 1 shows relative expression of Ucp1 and TET genes in response to ambient temperature changes. The authors suggest that among the 3 TET genes, Tet1 expression was highest in beige fat. However, these data compared expression at TN and cold relative to levels at RT and only the western blot for Tet1 was shown. To show clearly that Tet1 is the dominant TET gene expressed in beige fat, it is important that the authors complete this figure with the following:
 - a. Normalized absolute quantitation of Ucp1, Tet1, Tet2 and Tet3 transcript copies in the three conditions. Alternatively, show normalized transcript per million (TPM) values of these genes from RNA-seq data.
 - b. Show transcript (by qPCR) and protein (by Western blot) expression not only of Tet1, but also Tet2 and Tet3 in iWAT and BAT relative to other mouse tissues. Also, loss of Tet1 protein should be shown in the characterization of adipose-specific Tet1 KO (Supplementary Fig 3).
 - c. In Tet1 western blots, include the molecular size markers. Different protein isoforms have been characterized for TET1 (Zheng W. et al 2016 PMID 27916660). It is unclear which Tet1 isoform is shown here.
3. CpG methylation levels at gene promoters and enhancers of Ucp1 and Tet1, as determined by bisulfite conversion of DNA, in response to ambient temperature changes would be informative in Fig.1 (likely more than 5hmC-Seal in Supple. Fig. 9). At the least, base-resolution analysis of methylation will show whether the temperature-responsive changes in Ucp1 and Tet1 expression are associated or not with CpG methylation changes.
4. In Fig. 6F, the authors stated that interestingly, knocking down the Dnmts alone increased Ucp1 expression. However, the increase was very small and only approaching statistical significance. The authors need to show, in the first place, whether DNA methylation regulates Ucp1 expression during temperature changes.
5. It is unclear what OCR measurements the authors are calculating in Fig. 2I, 3I, 4F, 6E. What is oligomycin-resistant mitochondria respiration? If the authors are referring to ATP production, that should be oligomycin-sensitive (last rate measurement before oligomycin injection - minimum rate measurement after). What about the maximal respiration (maximum rate measurement after FCCP injection - non-mitochondrial respiration)?
6. Why did the authors not include the thermoneutral (TN) group in the RNAseq analysis?
7. All statistical analyses were performed using Student's t-test, which is not appropriate because all experiments shown involved more than 2 groups and one variable (treatment or time course and genotype). For example, one-way ANOVA should be used in Fig 1, two-way ANOVA in Fig. 2 with multiple testing correction.

Minor comments:

1. Page 2: Yoo et al., 2017 was cited but cannot be found in references.
2. Figure 2 legend. I,J,K should be G,H,I. Legend for Figure 3G,H,I is an exact copy of Figure 2G,H,I. Knockdown is confused for overexpression also in Supplementary Fig 2 legend.

3. What are eWat and pWat in Fig. 5H?

Reviewer #2 (Remarks to the Author):

The study done by Sneha et al entitled "TET1 is a beige adipocyte-selective epigenetic suppressor of thermogenesis" is interesting with nice data, but the current form needs to be improved by addressing several issues as below:

Major comments

#1. In the background, authors should have more paragraphs regarding characteristics, biology and function of brown and beige adipocytes, especially the brown and beige adipogenesis, author should reference following papers:

(1) Brown and brite adipocytes: Same function, but different origin and response. Chu DT, Gawronska-Kozak B. *Biochimie*. 2017 Jul;138:102-105. doi: 10.1016/j.biochi.2017.

(2) Human thermogenic adipocytes: a reflection on types of adipocyte, developmental origin, and potential application. Chu DT, Tao Y. *J Physiol Biochem*. 2017 Feb;73(1):1-4. doi: 10.1007/s13105-016-0536-y.

(3). OPA1 in Lipid Metabolism: Function of OPA1 in Lipolysis and Thermogenesis of Adipocytes. Chu DT, Tao Y, Taskén K. *Horm Metab Res*. 2017 Apr;49(4):276-285. doi: 10.1055/s-0043-100384.

#2. Authors stated that "expression data suggested that TET1 and TET2 have a functional role in the regulation of thermogenesis in beige fat, thus we sought to test the effects of their downregulation in beige adipocyte thermogenesis". I think this statement should be supported by more data than that they are presenting in the Figure 1, and Supplement Fig. 1, authors should have the data comparing the expression of Ucp1 and TETs among iWAT, iBAT and gonadal fat (or other visceral fats – for white fat depots). Beside, in vivo, I think it should be interesting to see the expression of Ucp1 and TET in pure white, brown and brite adipocytes in vitro, in this model they may differentiate the white, brown and brite adipocytes from adipose tissue stem cells or cell lines.

#3. In the experiment of gain- and loss-of-function of individual Tets using an immortalized beige cell line, did authors check the expression of Ppar alpha (Ppara)? Normally, the thermogenic function of brown and brite fat cells depend on 3 key genes including Ucp1, Pgc1a and Ppara. If not, I suggest they should include that data in the figure 2.

Minor comments

#1. Figure 1 E and F, in the legend, authors should indicate clearly they measure expression of TET1 in iWAT, eWAT, and BAT in which conditions (RI, TN or cold)?

#2. Authors should check again the western blot experiments; it seems that they did not load the same amount of protein in each well of gel, that is why they have the huge variation in the expression of internal control protein (GAPDH), e.g. Fig 4G.

#3. Authors should explain somewhere in the manuscript why did they use 1 uM forskolin (Forsk) or 1 uM norepinephrine (NE) for stimulating the thermogenesis in the beige adipocytes? Because on our hands the doses of Forsk and NE were so different to stimulating beige adipocytes in vitro

Reviewer #3 (Remarks to the Author):

In this manuscript Villivalam et al describe a role for the nuclear protein TET1 in thermogenesis, energy expenditure, and the regulation of body weight. The authors first show that TET1 expression is suppressed during cold exposure and in vitro by adrenergic agonists. In vitro loss and gain of function studies in beige adipocytes establish a functional anti-thermogenic role for TET1. In mice, aP2-cre mediated deletion of TET1 results in a profound hypermetabolic phenotype

which is characterized by increased whole body respiration, resistance to cold-induced hypothermia, resistance to diet-induced obesity, and improvements to glucose and insulin sensitivity. These physiologic changes are accompanied by a profound molecular “browning” of the inguinal white fat. Mechanistically, the authors show that TET1 demethylase activity is not required for its anti-thermogenic functions; instead, TET1 appears to interact with HDAC1 to regulate the transcription of adipose thermogenesis genes. Overall, I found the experiments themselves to be well-done and the writing to be easy to read.

Major comments: I have a serious concern about the main animal model used to support the claims of the manuscript. The authors use aP2-cre to delete TET1. aP2-cre, as the authors acknowledge, is well-known to exhibit recombination in multiple cell types beyond adipocytes including endothelial cells and certain neurons in the brain. Importantly, the authors show that the adiponectin-Cre cross, which is the gold standard in the field for adipose-specific deletion, did not alter TET1 levels in adipose tissues. My interpretation is that the effects observed in the aP2-cre::TET1 mouse model are not due adipose TET1, but rather TET1 deficiency in some other cell type. Consequently the data as presented, in my opinion, do not support the major conclusion that adipose TET1 suppresses thermogenesis. My enthusiasm with the manuscript in its current form is therefore quite limited. What would be far more convincing is for the authors to systematically and specifically knock out TET1 from cell types marked by aP2-cre, such as endothelial cells or certain neurons, to definitively establish the cell type driving the metabolic phenotypes.

Minor comments: Loss of TET1 protein in the in vivo models should be provided using the anti-TET1 antibody from Fig. 1.

We thank the reviewers for their insightful and largely positive comments along with their constructive criticism. We respond below to the general remarks of the editor and then separately to each specific concern raised. The comments of the reviewers are in bold, and our responses are in plain text. The modifications we made are highlighted in green in the revised manuscript.

Reviewer #1 (Remarks to the Author):

This is a well-written manuscript reporting that the DNA dioxygenase Tet1 is a thermosensitive repressor of the thermogenesis program in beige fat. Expression of Tet1 is suppressed by cold temperature, which then facilitates induction of key thermogenic genes. Using adipose-selective KO mice, the authors showed metabolic changes in KO mice that improved cold tolerance and increased energy expenditure. Interestingly, thermogenic genes were already up-regulated at RT in inguinal adipocytes of Tet1 KO mice, suggesting an epigenetic mechanism that enhances thermogenic gene expression during cold exposure when Tet1 is repressed. The authors further demonstrate that the gene suppressive role of Tet1 is DNA demethylase-independent and involves coordination with HDAC1. Overall, the study is interesting and proposes Tet1 as a potential therapeutic target to increase energy expenditure in obesity and related metabolic syndromes. However, there are some aspects of the study that need improvement.

Major comments:

1. The claim that TET1 is a powerful epigenetic regulator of the thermogenic gene program in beige adipocytes is over-stated. While the study shows an involvement of Tet1, there is no evidence that Tet1 is sufficient to repress thermogenesis in the absence of co-repressors such as HDAC1.

: We agree with your comment. We've switched "powerful" to "important".

2. Figure 1 shows relative expression of Ucp1 and TET genes in response to ambient temperature changes. The authors suggest that among the 3 TET genes, Tet1 expression was highest in beige fat. However, these data compared expression at TN and cold relative to levels at RT and only the western blot for Tet1 was shown. To show clearly that Tet1 is the dominant TET gene expressed in beige fat, it is important that the authors complete this figure with the following:

: We thank the reviewer for the comment. First, we need to clarify a misunderstanding. We stated that "*Tet1* expression was the highest in beige fat and at a minimum in iBAT". Here, we meant that *Tet1* expression is the highest in beige fat compared to eWAT and BAT and not that *Tet1* expression is highest compared to other Tets. *Tet1* did, however, have the highest degree of change in its expression between temperatures. With that said, we will still address the reviewer's suggestions as they are valuable questions.

a. Normalized absolute quantitation of Ucp1, Tet1, Tet2 and Tet3 transcript copies in the three conditions. Alternatively, show normalized transcript per million (TPM) values of these genes from RNA-seq data.

: To support our point on the higher expression of Tet1 in iWAT over eWAT and BAT, we performed digital PCR and provide the copy number (**Fig. 1E**).

b. Show transcript (by qPCR) and protein (by Western blot) expression not only of Tet1, but also Tet2 and Tet3 in iWAT and BAT relative to other mouse tissues. Also, loss of Tet1 protein should be shown in the characterization of adipose-specific Tet1 KO (Supplementary Figs. 3B, D).

: We now provide both qPCR and western blot data for all three Tets' expression in mouse tissues (**Figs. 1D, M**) and have added the data for the loss of Tet1 protein (**Supplemental Fig. 3**).

c. In Tet1 western blots, include the molecular size markers. Different protein isoforms have been characterized for TET1 (Zheng W. et al 2016 PMID 27916660). It is unclear which Tet1 isoform is shown here.

: Thank you for the comment. We've now added molecular size markers to the western blots. In fact, we were aware of this article, which shows Tet1 isoform switching from somatic Tet1 (Tet1s) to embryonic Tet1 (Tet1e) during early mouse embryonic development. However, we have not consistently detected the additional band corresponding to Tet1e as Zheng et al reported. We postulate that the isoform switching might be a phenomenon more common in embryonic development.

3. CpG methylation levels at gene promoters and enhancers of Ucp1 and Tet1, as determined by bisulfite conversion of DNA, in response to ambient temperate changes would be informative in Fig.1 (likely more than 5hmC-Seal in Supple. Fig. 9). At the least, base-resolution analysis of methylation will show whether the temperature-responsive changes in Ucp1 and Tet1 expression are associated or not with CpG methylation changes.

: We are grateful for the suggestion. Accordingly, we have conducted and now include bisulfite PCR to obtain base-pair resolution of methylation change in the three temperature conditions at the *Ucp1* and *Tet1* promoter and enhancer regions (**Supplemental Fig. 13**). We noted an overall reduction in CpG methylation at one of the *Ucp1* promoter regions at lower temperature conditions (**Supplemental Figs. 13A, E**). Other than that, no obvious changes were detected in the other tested *Ucp1* and *Tet1* promoter/enhancer regions (**Supplemental Figs. 13B-D, F-H, O-Q, R-T**).

In terms of methylation at individual CpGs, it is difficult to make any general conclusions as some of the CpG sites appear to be hypermethylated while some others are hypomethylated or have no correlation with lower temperature conditions (**Supplemental Figs. 13I-L, 13U-W**). Further investigation is warranted to understand the functional role of DNA methylation changes at individual CpG sites.

4. In Fig. 6F, the authors stated that interestingly, knocking down the Dnmts alone increased Ucp1 expression. However, the increase was very small and only approaching statistical significance. The authors need to show, in the first place, whether DNA methylation regulates Ucp1 expression during temperature changes.

: Thank you for the comment. As shown in **Supplemental Fig. 13**, one of the UCP1 promoter regions shows overall a reduced methylation upon cold exposure compared to thermoneutrality, correlating with UCP1 induction. Further studies are necessary to define loci specific DNA methylation in regulating UCP1 transcription.

5. It is unclear what OCR measurements the authors are calculating in Fig. 2I, 3I, 4F, 6E. What is oligomycin-resistant mitochondria respiration? If the authors are referring to ATP production, that should be oligomycin-sensitive (last rate measurement before oligomycin injection - minimum rate measurement after). What about the maximal respiration (maximum rate measurement after FCCP injection - non-mitochondrial respiration)?

: We appreciate the comment. To make clear our OCR data, we now present non-mitochondrial (OCR after adding Rotenone/AA), protein leak (by subtracting non-mitochondrial respiration from the oligomycin rate), ATP production (by subtracting the oligomycin rate from baseline cellular OCR), and maximum respiration (subtracting non-mitochondrial respiration from maximal OCR).

6. Why did the authors not include the thermoneutral (TN) group in the RNAseq analysis?

: This is because we did not see differences in energy expenditure and in the expression of key thermogenic genes in this condition.

7. All statistical analyses were performed using Student's t-test, which is not appropriate because all experiments shown involved more than 2 groups and one variable (treatment or time course and genotype). For example, one-way ANOVA should be used in Fig 1, two-way ANOVA in Fig. 2 with multiple testing correction.

: Thank you for the comments. Per suggestion, we now perform the appropriate statistics.

Minor comments:

1. Page 2: Yoo et al., 2017 was cited but cannot be found in references.

: Thank you for catching that. The corresponding reference was incorrect, and we have now fixed it.

2. Figure 2 legend. I,J,K should be G,H,I. Legend for Figure 3G,H,I is an exact copy of Figure 2G,H,I. Knockdown is confused for overexpression also in Supplementary Fig 2 legend.

: Thanks for catching this. We fixed the legends.

3. What are eWat and pWat in Fig. 5H?

: We added full descriptions in the legend for clarification (eWAT: epididymal WAT, pWAT: perirenal WAT).

Reviewer #2 (Remarks to the Author):

The study done by Sneha et al entitled "TET1 is a beige adipocyte-selective epigenetic suppressor of thermogenesis" is interesting with nice data, but the current form needs to be improved by addressing several issues as below:

Major comments

#1. In the background, authors should have more paragraphs regarding characteristics,

biology and function of brown and beige adipocytes, especially the brown and beige adipogenesis, author should reference following papers:

(1) Brown and brite adipocytes: Same function, but different origin and response. Chu DT, Gawronska-Kozak B. *Biochimie*. 2017 Jul;138:102-105. doi: 10.1016/j.biochi.2017.

(2) Human thermogenic adipocytes: a reflection on types of adipocyte, developmental origin, and potential application. Chu DT, Tao Y. *J Physiol Biochem*. 2017 Feb;73(1):1-4. doi: 10.1007/s13105-016-0536-y.

(3). OPA1 in Lipid Metabolism: Function of OPA1 in Lipolysis and Thermogenesis of Adipocytes. Chu DT, Tao Y, Taskén K. *Horm Metab Res*. 2017 Apr;49(4):276-285. doi: 10.1055/s-0043-100384.

: Thank you for the suggestion. We have added an additional paragraph in the introduction with the suggested references.

#2. Authors stated that “expression data suggested that TET1 and TET2 have a functional role in the regulation of thermogenesis in beige fat, thus we sought to test the effects of their downregulation in beige adipocyte thermogenesis”. I think this statement should be supported by more data than that they are presenting in the Figure 1, and Supplement Fig. 1, authors should have the data comparing the expression of Ucp1 and TETs among iWAT, iBAT and gonadal fat (or other visceral fats – for white fat depots). Beside, in vivo, I think it should be interesting to see the expression of Ucp1 and TET in pure white, brown and brite adipocytes in vitro, in this model they may differentiate the white, brown and brite adipocytes from adipose tissue stem cells or cell lines.

: We agree with the reviewer. We added more expression data on Tets. Per the suggestion, we added Tet expression in BAT and eWAT in **Fig. 1** and **Supplemental Fig. 1**. In addition, we also compared their expression levels in white (3T3-L1) and immortalized ‘beige’ and ‘brown’ preadipocytes and adipocytes (**Figs. 1J-M**).

#3. In the experiment of gain- and loss-of-function of individual Tets using an immortalized beige cell line, did authors check the expression of Ppar alpha (Ppara)? Normally, the thermogenic function of brown and brite fat cells depend on 3 key genes including Ucp1, Pgc1a and Ppara. If not, I suggest they should include that data in the figure 2.

: We checked *Ppara* expression in the *in vitro* studies. In our hands, we did not see a significant change after forskolin treatment even in the control cells (**Figs. 2E, 3E**). There was a trend toward a decrease in *Ppara* expression in Tet1 overexpressor (**Fig. 3E**), while *Tet1* knock-down did not have a dramatic impact on the basal or forskolin-stimulated *Ppara* expression (**Fig. 2E**). However, we noted that *Ppara* expression was increased in primary KO adipocytes compared to WT cells (**Figs. 4J-L**).

Minor comments

#1. Figure 1 E and F, in the legend, authors should indicate clearly they measure expression of TET1 in iWAT, eWAT, and BAT in which conditions (RT, TN or cold)?

: It was measured at RT. We have added this detail.

#2. Authors should check again the western blot experiments; it seems that they did not

load the same amount of protein in each well of gel, that is why they have the huge variation in the expression of internal control protein (GAPDH), e.g. Fig 4G.

: We have repeated the western blotting and replaced the loading control with B-ACTIN (Fig. 4G). Quantified signal density to B-ACTIN is also presented (Fig. 4G).

#3. Authors should explain somewhere in the manuscript why did they use 1 uM forskolin (Forsk) or 1 uM norepinephrine (NE) for stimulating the thermogenesis in the beige adipocytes? Because on our hands the doses of Forsk and NE were so different to stimulating beige adipocytes in vitro.

: We agree that the degree of inducing thermogenesis is different between the two reagents. We switched reagents for a practical reason. It was often difficult to see drug responses when using forskolin in the Seahorse experiment, and we had a better luck using NE.

Reviewer #3 (Remarks to the Author):

In this manuscript Villivalam et al describe a role for the nuclear protein TET1 in thermogenesis, energy expenditure, and the regulation of body weight. The authors first show that TET1 expression is suppressed during cold exposure and in vitro by adrenergic agonists. In vitro loss and gain of function studies in beige adipocytes establish a functional anti-thermogenic role for TET1. In mice, aP2-cre mediated deletion of TET1 results in a profound hypermetabolic phenotype which is characterized by increased whole body respiration, resistance to cold-induced hypothermia, resistance to diet-induced obesity, and improvements to glucose and insulin sensitivity. These physiologic changes are accompanied by a profound molecular “browning” of the inguinal white fat. Mechanistically, the authors show that TET1 demethylase activity is not required for its anti-thermogenic functions; instead, TET1 appears to interact with HDAC1 to regulate the transcription of adipose thermogenesis genes. Overall, I found the experiments themselves to be well-done and the writing to be easy to read.

Major comments: I have a serious concern about the main animal model used to support the claims of the manuscript. The authors use aP2-cre to delete TET1. aP2-cre, as the authors acknowledge, is well-known to exhibit recombination in multiple cell types beyond adipocytes including endothelial cells and certain neurons in the brain. Importantly, the authors show that the adiponectin-Cre cross, which is the gold standard in the field for adipose-specific deletion, did not alter TET1 levels in adipose tissues. My interpretation is that the effects observed in the aP2-cre::TET1 mouse model are not due adipose TET1, but rather TET1 deficiency in some other cell type. Consequently the data as presented, in my opinion, do not support the major conclusion that adipose TET1 suppresses thermogenesis. My enthusiasm with the manuscript in its current form is therefore quite limited. What would be far more convincing is for the authors to systematically and specifically knock out TET1 from cell types marked by aP2-cre, such as endothelial cells or certain neurons, to definitively establish the cell type driving the metabolic phenotypes.

We appreciate the concern. Coming from Dr. Evan Rosen's lab, which created adiponectin-Cre, we couldn't agree more with this reviewer's concern. Unfortunately, the *Tet1* allele was not

getting efficiently deleted using adiponectin-Cre mice in our hands (**Supplemental Fig. 3A, B**). To address the non-specificity issue using aP2-Cre, we've done additional studies. We believe that adipocytes are one of the main contributors to the thermogenesis phenotype in aP2-Tet1 KO mice, though there is still a possibility that there might be additional contributing factors from other cell types/tissue in this model. To support these notions, we provide the following results:

1. We noted a cell autonomous effect of Tet1 KD and OE on thermogenic gene regulation and mitochondrial respiration (**Figs. 2, 3**).
2. We achieved adipocyte-selective knock-down of *Tet1* by injecting Tet1f/f mice with AAV8-Cre, whose expression is driven by the human adiponectin promoter (Chella Krishnan et al., 2019; O'Neill et al., 2014). Of note, we delivered virus by directly injecting to iWAT to minimize off-target effect. We saw ~ 60% reduced expression of Tet1 in inguinal WAT but not in other tissues and this was sufficient to increase thermogenic gene expression and improve cold tolerance (**Supplemental. Fig. 8**).
3. We also investigated the possible contribution from macrophages since aP2-Cre is active in macrophages and it has been proposed that altered macrophage polarization can affect browning. Our FACS analysis did not find a major difference in M1 vs. M2 macrophage populations between WT and KO mice (**Supplemental Fig. 6A**). We also conducted a co-culture experiment using beige adipocytes and primary macrophages isolated from aP2-Tet1 KO vs WT mice. Again, no major differences were noted in the ability to induce thermogenic genes in beige adipocytes between genotypes (**Supplemental Figs. 6B, C**).
4. We provide negative data from *Tet1* KO mice generated using PDGFRa-Cre mice (**Supplemental. Fig. 7**). PDGFRa-Cre is often used to target the preadipocyte population but is expressed in a wide array of mesenchymal tissues, including the lung, heart, intestine, skin, and cranial facial mesenchyme (Boström et al., 1996; Chong et al., 2013; Karlsson et al., 1999; Karlsson et al., 2000; Lindahl et al., 1997; McCarthy et al., 2016). We found no remarkable phenotypic changes in energy expenditure, insulin sensitivity, glucose tolerance, or thermogenic gene expression in inguinal fat and BAT (**Supplemental. Fig. 7**).

Minor comments: **Loss of TET1 protein in the in vivo models should be provided using the anti-TET1 antibody from Fig. 1.**

Thank you for pointing that out. We have now added protein analysis (**Supplemental. Fig. 3**).

References

1. Chella Krishnan, K., Sabir, S., Shum, M., Meng, Y., Acín-Pérez, R., Lang, J. M., Floyd, R. R., Vergnes, L., Seldin, M. M., Fuqua, B. K., Jayasekera, D. W., Nand, S. K., Anum, D. C., Pan, C., Stiles, L., Péterfy, M., Reue, K., Liesa, M., & Lusis, A. J. (2019). Sex-specific metabolic functions of adipose Lipocalin-2. *Molecular Metabolism*, 30, 30–47. <https://doi.org/10.1016/j.molmet.2019.09.009>

2. O'Neill, S. M., Hinkle, C., Chen, S. J., Sandhu, A., Hovhannisyan, R., Stephan, S., Lagor, W. R., Ahima, R. S., Johnston, J. C., & Reilly, M. P. (2014). Targeting adipose tissue via systemic gene therapy. *Gene Therapy*, 21(7), 653–661. <https://doi.org/10.1038/gt.2014.38>

REVIEWERS' COMMENTS second round:

Reviewer #1 (Remarks to the Author):

The authors have improved the manuscript with additional data to support their interesting findings. Most of the concerns raised during the review have been addressed, with only these minor issues with some of the new data:

1. Figure 1. I appreciate the additional Western blots to demonstrate TET protein expression in various tissues and cell lines. In the previous version, it was unclear to me whether Tet1 is the key player among the three TETs in thermogenic regulation because it is the dominant TET protein expressed in iWAT or because each TET functions differently in temperature response. The Western blot in Fig 1D is detecting all three TETs in iWAT and eWAT, suggesting that all three TETs may indeed be moderately expressed in these adipose tissues. However, the new digital PCR data measuring copy number of Tet1 in fat tissues is not adding much information (and may be better off removed) without normalization to a housekeeping transcript and comparison with Tet2 and Tet3 copy numbers. The figure also appears incomplete without western blots for Fig 1E (Ucp1), 1G (Tet2) and 1H (Tet3), as has been provided for Fig 1F (Tet1). Moreover, relative expression of Tet2 and Tet3 in primary beige adipocytes in response to Forsk and NE (Fig 1N-O) could be shown. While these missing data should not impede acceptance of this manuscript, it should be feasible for the authors to add these data before publication.

2. Supplemental Fig 13. Bisulfite sequencing at Ucp1 and Tet1. Please provide positional coordinates (in bp) of amplicon ends relative to TSS and/or location of BS loci in a gene schematic to clarify whether those regions, especially promoters, are within or flanking CpG islands. Are PCR duplicate (identical reads) sequences removed in the analysis? The clonal sequences suggest either almost fully methylated or unmethylated regions, which may result from detection of different cell types with or without expression in the tissues. I agree that it is difficult to make any general conclusions based on hypermethylation or hypomethylation of individual CpGs. It suffices to say that there are no notable changes at those loci examined, except for a small reduction in CpG methylation at Ucp1_P1 in response to cold, in line with gene expression changes being largely independent of DNA methylation status or Tet1 catalytic activity.

Letterings for figure panels are wrong in several figure legends.

Fig 1J-L. Labels for Tet1, Tet2 and Tet3 missing.

Fig 3I. WT should be +NE?

Fig 7F-H are missing.

Sup Fig 3D. Please add labels for Tet1 and loading control.

Reviewer #2 (Remarks to the Author):

My comments and questions have been addressed, and the manuscript has been improved

Reviewer #3 (Remarks to the Author):

The authors have provided sufficient additional experimental data to address my concerns.

It would be appropriate for the authors to modify the abstract of this manuscript to explicitly state that they used the fabp4-cre driver to generate the TET1-deficient model that they are studying here.

REVIEWERS' COMMENTS:

We thank reviewers for additional comments. We respond below to the specific questions raised. The comments of the reviewers are in bold, and our responses are in plain text.

Reviewer #1 (Remarks to the Author):

The authors have improved the manuscript with additional data to support their interesting findings. Most of the concerns raised during the review have been addressed, with only these minor issues with some of the new data:

1. Figure 1. I appreciate the additional Western blots to demonstrate TET protein expression in various tissues and cell lines. In the previous version, it was unclear to me whether Tet1 is the key player among the three TETs in thermogenic regulation because it is the dominant TET protein expressed in iWAT or because each TET functions differently in temperature response. The Western blot in Fig 1D is detecting all three TETs in iWAT and eWAT, suggesting that all three TETs may indeed be moderately expressed in these adipose tissues. However, the new digital PCR data measuring copy number of Tet1 in fat tissues is not adding much information (and may be better off removed) without normalization to a housekeeping transcript and comparison with Tet2 and Tet3 copy numbers. The figure also appears incomplete without western blots for Fig 1E (Ucp1), 1G (Tet2) and 1H (Tet3), as has been provided for Fig 1F (Tet1). Moreover, relative expression of Tet2 and Tet3 in primary beige adipocytes in response to Forsk and NE (Fig 1N-O) could be shown. While these missing data should not impede acceptance of this manuscript, it should be feasible for the authors to add these data before publication.

: Thank you very much for your confirmation and comments. We were able to manage to do additional western blotting for Figs 1E-H. However, we are not able to do additional western blotting for Figs 1N-O as our tissue culture work is limited due to COVID-19.

2. Supplemental Fig 13. Bisulfite sequencing at Ucp1 and Tet1. Please provide positional coordinates (in bp) of amplicon ends relative to TSS and/or location of BS loci in a gene schematic to clarify whether those regions, especially promoters, are within or flanking CpG islands. Are PCR duplicate (identical reads) sequences removed in the analysis? The clonal sequences suggest either almost fully methylated or unmethylated regions, which may result from detection of different cell types with or without expression in the tissues. I agree that it is difficult to make any general conclusions based on hypermethylation or hypomethylation of individual CpGs. It suffices to say that there are no notable changes at those loci examined, except for a small reduction in CpG methylation at Ucp1_P1 in response to cold, in line with gene expression changes being largely independent of DNA methylation status or Tet1 catalytic activity.

: We now added coordinates info regarding CpG flanking regions in Sup Fig. 13. To answer your question about bisulfite PCR study, bisulfite-treated DNA was amplified and subcloned to T-vector. Circular map corresponds to CpG methylation sequenced from miniprep DNA of each clone and so, no PCR duplicate issues are involved in data analysis.

Letterings for figure panels are wrong in several figure legends.

Fig 1J-L. Labels for Tet1, Tet2 and Tet3 missing.

Fig 3I. WT should be +NE?

Fig 7F-H are missing.

Sup Fig 3D. Please add labels for Tet1 and loading control.

: We are sorry about the mistakes and thank you for catching them. Now, we corrected all.

Reviewer #2 (Remarks to the Author):

My comments and questions have been addressed, and the manuscript has been improved.

: We appreciate the comments.

Reviewer #3 (Remarks to the Author):

The authors have provided sufficient additional experimental data to address my concerns.

It would be appropriate for the authors to modify the abstract of this manuscript to explicitly state that they used the fabp4-cre driver to generate the TET1-deficient model that they are studying here.

: Thank the reviewer for the suggestion. We fully agree with reviewer's suggestion and now we specify that by saying "Adipose-selective Tet1 knockout mice generated by using Fabp4-Cre improves cold tolerance and increases energy expenditure and protects against diet-induced obesity and insulin resistance."